# The K219T-Lamin mutation induces conduction defects through epigenetic inhibition of *SCN5A* in human cardiac laminopathy

Nicolò Salvarani [1,2,8], Silvia Crasto[1,2,8], Michele Miragoli [1,2,3], Alessandro Bertero [4], Marianna Paulis[1,2], Paolo Kunderfranco [2], Simone Serio[2], Alberto Forni[5], Carla Lucarelli[5], Matteo Dal Ferro[6], Veronica Larcher[2], Gianfranco Sinagra [6], Paolo Vezzoni[1,2], Charles E. Murry [4], Giuseppe Faggian[5], Gianluigi Condorelli [1,2,7] & Elisa Di Pasquale[1,2]

Mutations in *LMNA*, which encodes the nuclear proteins Lamin A/C, can cause cardio-myopathy and conduction disorders. Here, we employ induced pluripotent stem cells (iPSCs) generated from human cells carrying heterozygous K219T mutation on *LMNA* to develop a disease model. Cardiomyocytes differentiated from these iPSCs, and which thus carry K219T-*LMNA*, have altered action potential, reduced peak sodium current and diminished conduction velocity. Moreover, they have significantly downregulated Na$_v$1.5 channel expression and increased binding of Lamin A/C to the promoter of *SCN5A*, the channel's gene. Coherently, binding of the Polycomb Repressive Complex 2 (PRC2) protein SUZ12 and deposition of the repressive histone mark H3K27me3 are increased at *SCN5A*. CRISPR/Cas9-mediated correction of the mutation re-establishes sodium current density and *SCN5A* expression. Thus, K219T-*LMNA* cooperates with PRC2 in downregulating *SCN5A*, leading to decreased sodium current density and slower conduction velocity. This mechanism may underlie the conduction abnormalities associated with LMNA-cardiomyopathy.

[1] Institute of Genetic and Biomedical Research (IRGB), UOS of Milan, National Research Council of Italy, Milan 20138, Italy. [2] Department of Cardiovascular Medicine and Laboratory of Medical Biotechnology, Humanitas Clinical and Research Center – IRCCS, Rozzano (MI) 20089, Italy. [3] Department of Medicine and Surgery, University of Parma, Parma 43121, Italy. [4] Institute for Stem Cell and Regenerative Medicine, University of Washington, Seattle 98109 WA, USA. [5] Division of Cardiac Surgery, University of Verona, Verona 37129, Italy. [6] Cardiovascular Department, "Ospedali Riuniti" and University of Trieste, Trieste 34129, Italy. [7] Humanitas University, Rozzano (MI) 20089, Italy. [8] These authors contributed equally: Nicolò Salvarani, Silvia Crasto. Correspondence and requests for materials should be addressed to G.C. (email: gianluigi.condorelli@hunimed.eu) or to E.D.P. (email: elisa.dipasquale@irgb.cnr.it)

C ardiomyopathy (CMP) is the most prevalent phenotype of laminopathies, a group of diseases caused by mutations of the nuclear lamina proteins Lamin A and C (Lamin A/C), which are encoded by *LMNA*. Lamin A/C are main components of the nuclear lamina, and play a key role in maintaining the structural integrity of the nucleus and in processes such as DNA replication, transcription and chromatin organization[1–4]. More than 200 heterozygous mutations on *LMNA* are known to cause over 20 phenotypes in which either tissues and organs are selectively affected—mainly bone, cartilage, skeletal muscle, and the heart—or multiple organ systems are involved, causing accelerated ageing (progeria)[5]. Cardiac defects eventually occur in all symptomatic patients, manifesting either as an isolated phenotype or in association with a variety of skeletal muscular dystrophies, such as Emery Dreifuss muscular dystrophy, limb girdle muscular dystrophy, and lipodystrophies[6].

At the clinical level, LMNA-dependent CMP (LMNA-CMP) is characterized by chamber dilatation, typically preceded by conduction defects and arrhythmic events, which may occur before any detectable dysfunction[7,8]. Notably, fatal arrhythmias and consequential sudden cardiac death occur in almost 50% of LMNA-CMP patients[9,10]. Although age at disease onset is variable, almost all affected patients become symptomatic by the sixth decade and eventually progress onto heart failure necessitating transplantation[11,12].

In vivo studies on *Lmna* knock-in and knock-out murine models have been fundamental in elucidating the effects of Lamin A/C deficiency or mutations on cardiac function. They reproduce the main clinical features of the human disease (i.e., dilated CMP and atrio-ventricular block), providing important pathophysiological insights[13–16]. However, while murine models are necessary for functional studies on whole heart, rodent and human cardiac physiologies differ significantly. Thus, mechanisms specific to the human species may be missed. Unfortunately, studies on human cells carrying *LMNA* mutations have been mostly limited to fibroblasts, skeletal muscle cells and adipocytes, while those on isolated cardiomyocytes (CMs) are scarce[17–19].

CMs differentiated from induced pluripotent stem cells (iPSCs) have been used to understand the mechanisms and pathophysiology of many cardiac diseases and provide a reliable tool for drug screening applications[20–22]. To date, very few studies investigating CMs derived from patient-specific iPSC lines carrying a mutation in *LMNA* (i.e., the R225X nonsense and R190W missense mutations) have been published: results indicated an effect on cellular senescence and apoptosis dependent on the ERK1/2 pathway in the presence of the nonsense mutation;[23,24] abnormal activation of ERK1/2 signaling and sarcomeric disorganization were reported in iPSC-CMs carrying the R190W mutation[25]. Nevertheless, our understanding of the pathophysiological mechanisms in LMNA-CMP remains incomplete, particularly with regard to conduction defects.

For the present study, we integrated functional and molecular investigations on CMs generated through iPSC technology to determine the effect of the K219T *LMNA* mutation[26] on the electrophysiological properties of CMs. CRISPR/Cas9-mediated gene-editing and ex-vivo analyses on heart specimens from patients with the K219T mutation were also employed. We found that this mutation specifically affects expression of the sodium channel $Na_v1.5$ via an epigenetic mechanism involving interplay with Polycomb Group proteins and targeting of the expression of the gene encoding this channel, *SCN5A*. Thus, we identify a cause–effect link between a Lamin A/C mutation, altered $Na_v1.5$ expression and CM action potential changes, which might explain the arrhythmogenic conduction defects in these patients.

## Results

### Generation of an iPSC-based model of LMNA-CMP.
We generated a model of cardiac laminopathy from iPSCs obtained from fibroblasts harvested from three members of a large family affected by the disease. The family, previously identified by us to carry the K219T mutation on *LMNA*[26], presented with dilated cardiomyopathy associated with arrhythmogenic conduction disorders. Detailed clinical information is provided in Supplementary Table 1, Supplementary Fig. 1, and Supplementary Extended Methods. Control iPSC lines were generated from two healthy family members not carrying the mutation.

Following characterization of pluripotent status and genomic stability (Supplementary Figs. 2, 3), patient-derived and control iPSC lines (two clones per individual) were differentiated towards the cardiac lineage (iPSC-CMs) with a standard protocol[27] that reproducibly generates >80% pure CMs (Supplementary Fig. 4A). iPSC-CMs had spontaneous depolarizations and contractions (Fig. 1a, Supplementary Movies 1, 2), and expressed the cardiac-specific marker α-sarcomeric actinin (Supplementary Fig. 4A, B), which was, however, abnormally distributed in CMs differentiated from K219T-LMNA-iPSC lines (K219T-CMs): the staining was punctuated and fragmented, suggestive of the presence of disorganized myofilament structures (Supplementary Fig. 4B). This phenotype was observed in 50% of K219T-CMs, but in <15% control cells (CNTR-CMs; Supplementary Fig. 4C). Furthermore, measuring passive CM properties by patch clamp revealed a higher membrane capacity in K219T-CMs, indicating increased cell size (Supplementary Fig. 4D). Surprisingly, in contrast with previous observations by us and others on primary cells and mature adult CMs from affected patients[5,26,28], morphological analyses of K219T-CMs did not reveal any sign of the nuclear defects typical of laminopathic cells, such as nuclear blebbing and invagination (Supplementary Fig. 4E), characteristics which probably develop in adult CMs after long-term stress.

Thus, CMs generated from K219T-LMNA-iPSCs recapitulate some of the fundamental defects observed in LMNA-CMP, including dilatation and sarcomeric disorganization, and provide a suitable platform for investigating the molecular mechanisms underlying the disease.

### K219T-LMNA alters the action potential properties of CMs.
We then investigated how K219T-LMNA affects the functionality of CMs through comprehensive electrophysiological analysis of the action potential (AP) properties of iPSC-CMs with conventional patch-clamp methods.

We found that maximal diastolic potential (MDP) and maximal upstroke velocity ($dV/dt_{max}$), together with the related AP properties peak voltage (overshoot) and AP amplitude (APA), were significantly altered in K219T-CMs. Indeed, MDPs were significantly more depolarized in K219T-CM vs. CNTR-CMs (Fig. 1b—left panel, Supplementary Fig. 5). As a consequence of the observed depolarized MDP values, 73% of the analysed CNTR-CMs (33 cells out of 45) and 52% of K219T-CMs (24 out of 46 cells) exhibited spontaneous APs (Fig. 1a), which are never observed in healthy human adult ventricular CMs[29].

While AP duration was not changed in spontaneously active K219T-CMs, all the other analysed AP parameters—specifically $dV/dt_{max}$, overshoot and APA—were significantly decreased in K219T-CMs (Fig. 1c, d, Supplementary Table 2). In particular, $dV/dt_{max}$, which reflects the peak $Na^+$ current ($I_{Na}$) during the upstroke[30], was reduced by 36.1% in K219T-CMs. Of note, $dV/dt_{max}$ and its related AP properties (overshoot and APA) are a function of the electrical excitability of a CM and depend on the sarcolemmal fast voltage-dependent $Na^+$ channel ($Na_v1.5$) repertoire and its functional expression[31].

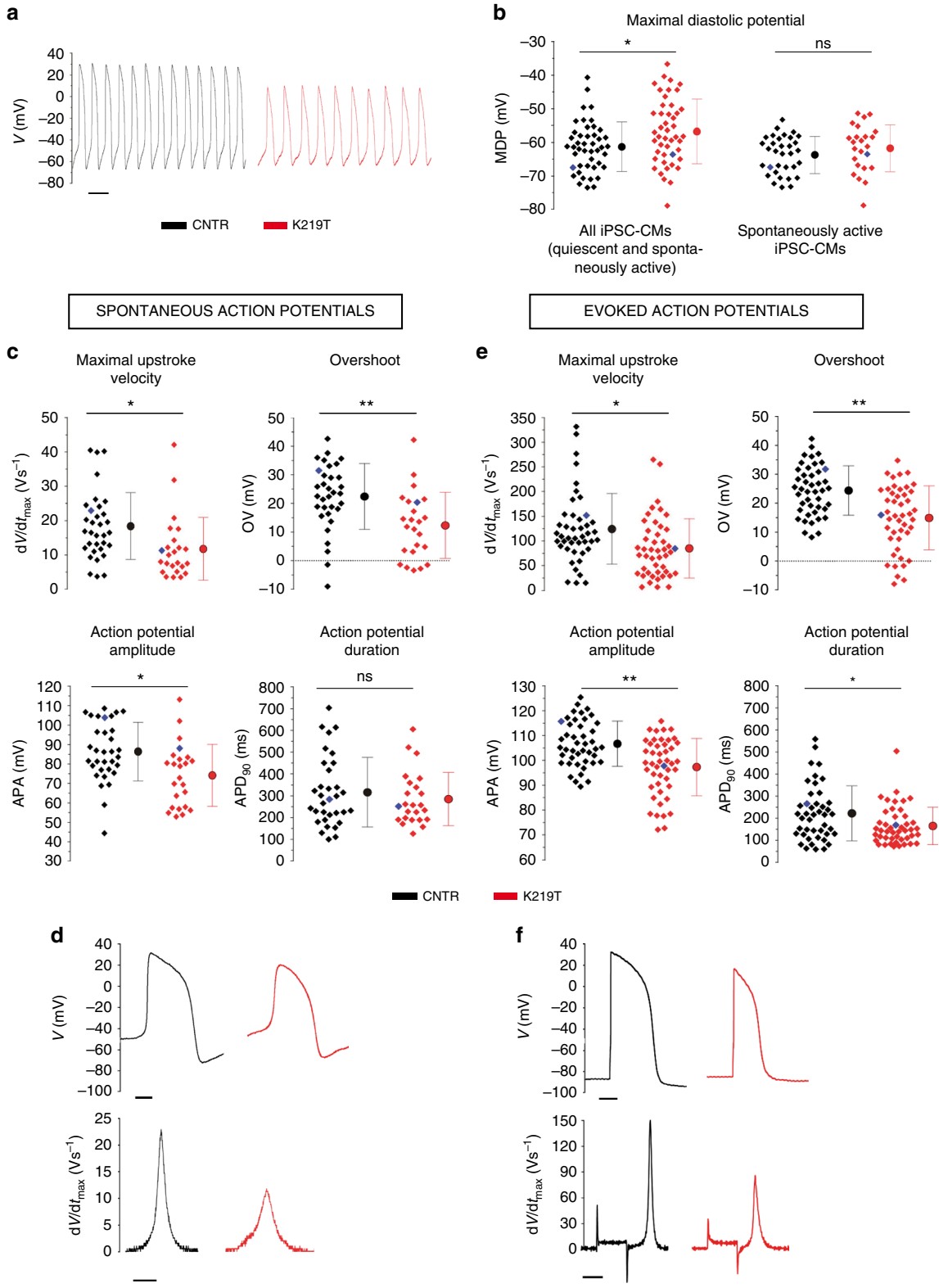

Since changes in MDP influence Na$^+$ channel availability, with the majority of channels in their inactivated state being at a more positive potential, we also studied the decreased excitability by recording AP parameters in CMs in which the MDP was adjusted to a physiological resting cardiac membrane potential range (i.e., ~ −80 mV) by electrical injection of a constant hyperpolarizing current —referred to here as evoked APs. This condition better resembles that of human adult ventricular CMs[32] and bypasses both the inactivation of Na$^+$ channels and mechanisms of spontaneous membrane depolarization[33]. As expected, we found higher values of d$V$/d$t_{max}$ and APA in iPSC-derived CMs with evoked APs vs spontaneously active CMs; again, AP properties related to excitability were significantly decreased in such electrically conditioned K219T-CMs (Fig. 1e, f, Supplementary Table 2, Supplementary Fig. 6).

Those data were confirmed when AP parameters were recorded in working-like CMs, selected on the basis of their AP

**Fig. 1** Electrophysiological properties of single iPSC-CMs. **a** Examples of spontaneous action potentials recorded in CNTR-CMs and K219T-CMs (scale bar, 1 sec). **b** Dot plot with maximal diastolic potential (MDP) data recorded from CNTR-CMs and K219T-CMs, relative to all iPSC-CMs (quiescent and spontaneously active cells: $-61.3 \pm 7.4$ mV vs. $-56.8 \pm 9.6$ mV—left panel) or to the spontaneously active cell population only (right panel), showing a significant depolarization of K219T-CMs vs. CNTR-CMs in the first condition. CNTR-CMs: $n = 45$; K219T-CMs: $n = 46$. **c**, **d** Patch clamp analysis of AP properties in spontaneously active iPSC-CMs. (CNTR: $n = 33$; K219T: $n = 24$). Dot plots for the main excitability features measured from spontaneous action potentials: maximal upstroke velocity ($dV/dt_{max}$), overshoot (OV), action potential amplitude (APA) and action potential duration at 90% repolarization (APD$_{90}$) are shown in **c**. Representative spontaneous action potentials (left; scale bar, 100 ms) and dV/dt (right; scale bar, 1 ms) in **d** (CNTR: $dV/dt_{max} = 18.4 \pm 9.7$ Vs$^{-1}$, $n = 33$; overshoot $= 22.6 \pm 11.2$ mV, $n = 33$; APA $= 86.4 \pm 15.2$ mV, $n = 33$. K219T-CMs: $dV/dt_{max} = 11.8 \pm 9.2$ V s$^{-1}$, $n = 24$; overshoot $= 12.3 \pm 11.6$ mV, $n = 24$; APA $= 74.2 \pm 16.1$ mV, $n = 24$). **e**, **f** Patch clamp analysis of AP properties in iPSC-CMs electrically adjusted to $-82$ mV (evoked APs-CNTR: $n = 45$; K219T: $n = 46$). Dot plots for the main excitability features measured from evoked action potentials: maximal upstroke velocity ($dV/dt_{max}$), overshoot (OV), action potential amplitude (APA) and action potential duration at 90% repolarization (APD$_{90}$) are shown in **e**. Representative evoked action potentials (left; scale bar, 100 ms) and dV/dt (right; scale bar, 2 ms) in **f**. Measurements highlighted with blue symbols in **c** and **e** were obtained from action potentials shown in **d** and **f**. All values are reported as mean ± SD. *$p < 0.05$; **$p < 0.005$ (unpaired $t$-test). Data are relative to CMs differentiated from 2 lines from each subjects (2 CNTR and 3 K219T-LMNA)

characteristics[21]. These additional evaluations were included in the first set of analyses to exclude potential underestimation of $I_{Na}$ due to contamination with pacemaker cells, which are more depolarized and devoid of Na$_v$1.5. Indeed, differentiation of iPSCs into CMs is known to give rise to a mixed population of atrial-like, ventricular-like, and pacemaker-like CMs, with the majority presenting the ventricular phenotype:[21,34] these different populations are typified by distinct AP parameters that reflect on AP morphology and determine specific functional outcomes[35]. In line with the results obtained in spontaneously active iPSC-CMs and in those electrically adjusted to $-82$ mV, AP parameters were significantly decreased in working-like K219T-CMs vs. CNTR-CMs (Supplementary Table 3), with a decrement in $dV/dt_{max}$ of 29.8% (Supplementary Fig. 6).

Collectively, these results indicate that K219T is associated with MDP depolarization and alterations in the electrical excitability profile, which were linked to a decrease in the fast $I_{Na}$ during the upstroke.

**K219T-CMs have reduced sodium current density**. To corroborate the previous findings and quantify the effects of K219T on $I_{Na}$ density in iPSC-CMs, we measured peak $I_{Na}$ density of fast voltage-dependent sodium channels (Na$_v$1.5) by whole-cell patch clamp, using specific voltage protocols and external ionic solutions. We found that average peak $I_{Na}$ density was significantly lower in K219T-CMs vs. CNTR-CMs (Fig. 2a), with a reduction of 41.3% at the membrane potential of $-30$ mV (Fig. 2b). Furthermore, no significant differences in voltage dependence of activation and inactivation parameters (half-maximal voltage [$V_{1/2}$] and slope factor [$k$]) were detected between the two (Fig. 2c, Supplementary Table 4).

Of note, Na$_v$1.5 currents and their voltage dependencies measured in CNTR-CMs were in line with published values obtained in control human iPSC-CMs[36,37], further proving the validity of our experimental setting and results.

Taken together, these results indicated a profound impairment of $I_{Na}$ in CMs carrying the K219T mutation, linked to the lower $dV/dt_{max}$ described above.

In addition, R190W—another heterozygous mutation on exon 3 of the gene, and also located in the rod domain of the protein (Supplementary Fig. 7A)—affected $I_{Na}$ in a similar manner to K219T (Supplementary Fig. 7B–D), indicating that the phenotype observed in K219T-CMs is not restricted to this specific mutation and suggesting a more general mechanism through which missense *LMNA* mutations may alter AP properties in affected CMs.

**K219T-Lamin A/C blunts cardiac impulse propagation**. Since $I_{Na}$ and MDP are both responsible for sustaining physiological cardiac impulse propagation across the myocardium[38], we next determined the consequences of the biophysical changes described above on cardiac conduction velocity. We thus measured optical AP propagation in a confluent multi-cellular culture setting consisting of 80-μm wide strands of electrically coupled iPSC-CMs stained with the voltage-sensitive dye DI-8-ANEPPS (Fig. 3a). K219T-CM strands failed to respond to increasing electrical stimulation (1 and 2 Hz), resulting in a 2:1, 3:1 or complete block of impulse propagation, possibly due to the prolonged absolute refractory period secondary to the reduced $I_{Na}$ (Fig. 3b, c)[39]. Furthermore, conduction velocities (CVs) in K219T-CM strands were significantly decreased (46.1%) (Fig. 3d), further supporting the notion of an impairment of impulse propagation in laminopathic cells.

Ablation of conduction, induced by the selective blockage of Na$_v$1.5 with tetrodotoxin (TTX), confirmed that the conduction defects observed in our cardiac laminopathy model are specifically driven by Na$_v$1.5 (Supplementary Fig. 8).

**Reduced Na$^+$ current is caused by downregulation of *SCN5A***. Regulation of transcription and chromatin organization are among the diverse roles attributed to Lamin A/C proteins[18,40–44]. We therefore hypothesized that mutation of Lamin A/C could interfere with the transcriptional program in K219T-CMs by perturbing the correct expression of Na$_v$1.5, leading to the conduction defects in our cellular model of laminopathy.

To prove this, we first analysed the expression of the Na$_v$1.5 encoding gene, *SCN5A*, with RT-qPCR. We found that the level of its transcript was significantly lower in K219T-CMs vs. CNTR-CMs (Fig. 4a). Assessment of protein expression (through Western blot and immunofluorescence analyses) confirmed reduced expression of Na$_v$1.5 in K219T-CMs (Fig. 4b–d). Similarly, we found a reduced level of *SCN5A* mRNA also in CMs carrying the R190W mutation (Supplementary Fig. 7E). These findings support the hypothesis that reduction of Na$^+$ currents in these cells is due to an effect on transcription by the Lamin A/C mutation.

**Epigenetic modulation of *SCN5A* expression by Lamin A/C**. To determine how the mutated Lamin A/C affects expression of *SCN5A*, we performed chromatin immunoprecipitation (ChIP) experiments to assess the occupancy of Lamin A/C on the gene's promoter. Binding of Lamin A/C to the *SCN5A* promoter was increased in K219T-CMs vs. CNTR-CMs (Fig. 5a); no enrichment was detected at the promoter of *JUN* (AP-1 Transcription factor subunit), used as a control, demonstrating the specificity of Lamin A/C immunoprecipitation. Instead Lamin A/C was equally enriched in CNTR-CMs and K219T-CMs at the transcription

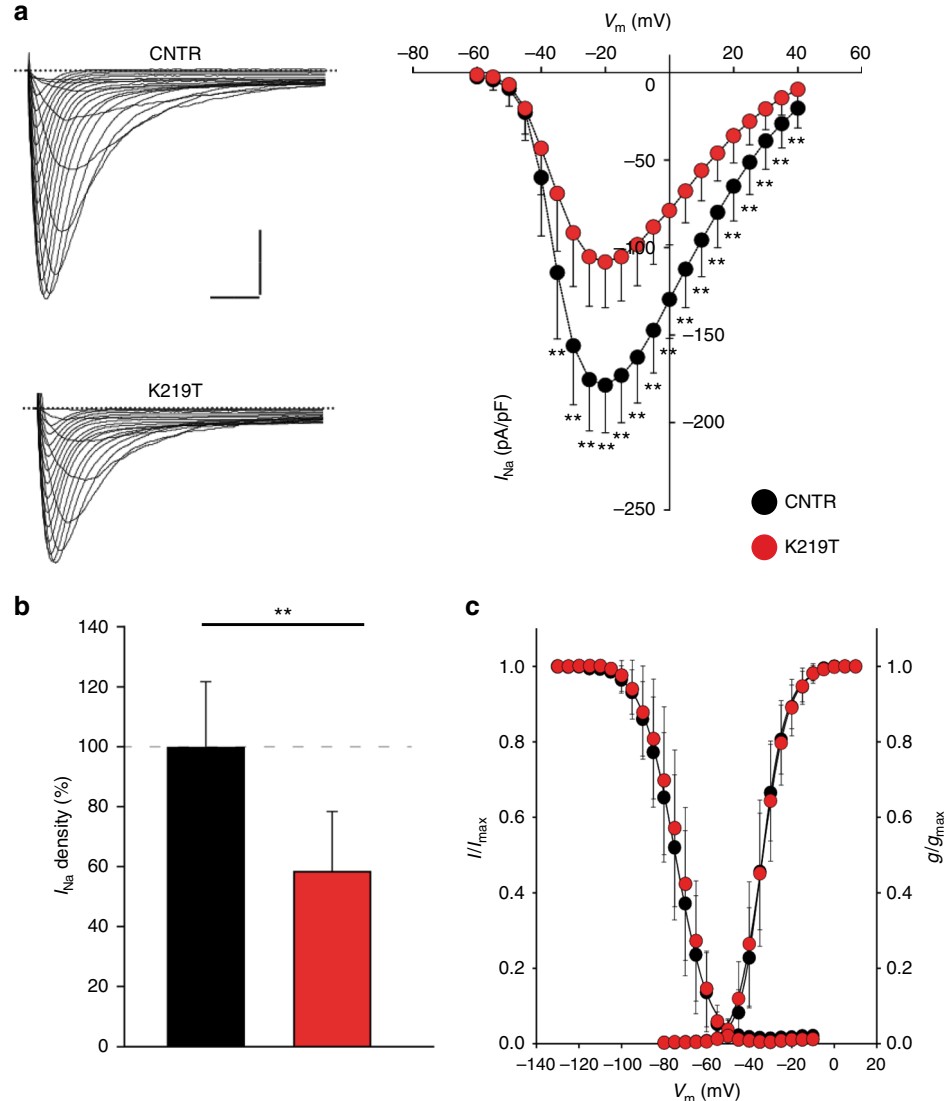

**Fig. 2** Voltage-gated sodium currents in CMs from K219T-iPSC lines. **a** Left panel: examples of Na$^+$ current ($I_{Na}$) traces recorded in CNTR- (top) and K219T- (bottom) CMs (Scale bar, 2 ms, 50 pA/pF). Right panel: $I$-$V$ curves constructed from average peak sodium current density as a function of voltage command measured in CNTR-CMs and K219T-CMs, showing a significant reduction in the latter. CNTR-CMs: $n = 23$; K219T-CMs: $n = 22$. **b** $I_{Na}$ density, measured at $-30$ mV, in K219T-CMs (91.58 ± 30.69 pA/pF) relative to CNTR-CMs (156.12 ± 33.73 pA/pF), expressed as a percentage. **c** Voltage dependences: steady state activation (CNTR-CMs: $n = 23$; K219T-CMs: $n = 22$)/inactivation curves (CNTR-CMs: $n = 18$; K219T-CMs: $n = 17$). All values are reported as mean ± SD. **p < 0.005 (unpaired t-test). Data are relative to CMs differentiated from 2 independent lines from 2 CNTR and 3 K219T-LMNA subjects

start sites of *SCN10A*, *DEFA3*, and *DEFA4* (Fig. 5a), known gene targets of Lamin A/C[18]. This finding excluded the possibility of an increased affinity of the antibody for K219T-Lamin A/C, strengthening our conclusion that the mutated protein is involved in regulating expression of *SCN5A* gene.

To further corroborate the specificity of Lamin A/C for *SCN5A*, we determined the expression of other genes encoding proteins playing key roles in cardiac conduction (i.e., *CX40*, *CX43*, *TBX3* and *TBX5*) and assessed Lamin A/C binding at their promoters. There were no significant changes in the expression of those genes in K219T-CMs vs. CNTR-CMs, and no differences in Lamin A/C enrichment at their promoters (Supplementary Fig. 9A, B), further confirming the major contribution of *SCN5A* regulation by Lamin A/C in determining the conduction phenotype.

To correlate the presence of Lamin A/C with the epigenetic landscape at the *SCN5A* locus in K219T-CMs, we performed a

ChIP analysis of two histone marks indicative of transcriptional repression—tri-methylated lysine 27 on histone 3 (H3K27me3) and tri-methylated lysine 9 on histone 3 (H3K9me3)—and one mark of transcriptional activation—tri-methylated lysine 4 on histone 3 (H3K4me3). The two repressive marks were enriched at the locus in K219T-CMs (Fig. 5b, Supplementary Fig. 10A); in contrast, H3K4me3 was enriched in CNTR-CMs (Supplementary Fig. 10B). These results support the hypothesis that diminished $I_{Na}$ density in K219T-CMs is due to reduced *SCN5A* expression which, in turn, is mediated by enrichment on its promoter of Lamin A/C and the two repressive histone marks H3K9me3 and, particularly, H3K27me3.

Nuclear lamins are also implicated in tethering chromatin to the nuclear lamina[45]; these interactions occur through large Lamin-Associated Domains (LADs) in chromatin and are predominantly associated with transcriptional silencing[42,46]. LADs are mostly cell-specific and may be distinguished into

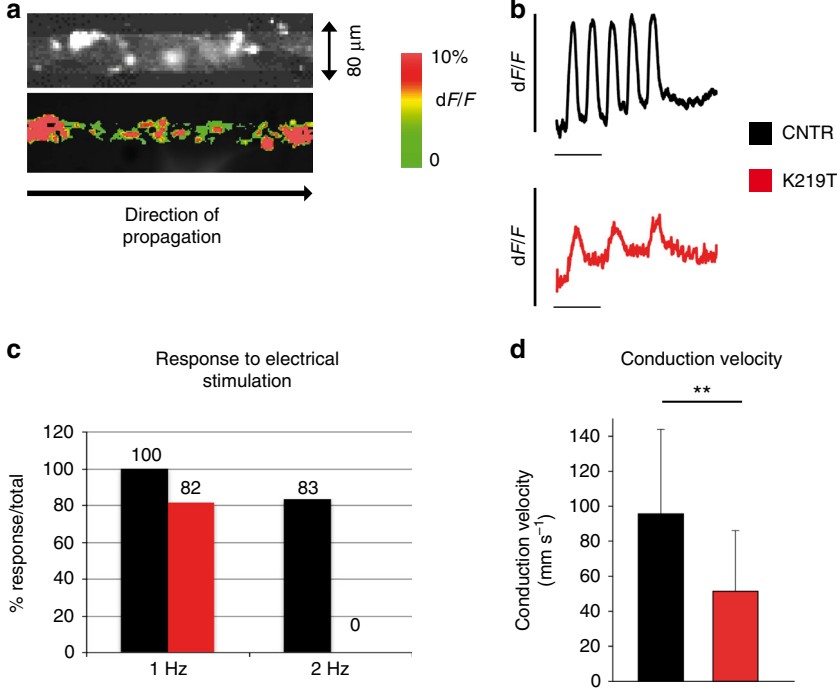

**Fig. 3** Generation and propagation of optical action potentials in K219T-CMs. **a** Representative images of a monolayer of iPSC-CMs in geometrically defined strands (80-μm wide, 1-cm long). Top: example of phase contrast image of a portion of a strand observed from the field of view; Bottom: same strand after complete activation following electrical stimulation. Strands were stimulated from the left-hand side. **b** Stimulated optical action potentials in a monolayer of CNTR- (top, black) and K219T- (bottom, red) CMs, showing that the latter fail to respond to a stimulation of 2 Hz. **c** Percentage of propagated action potentials at 1 and 2 Hz stimulation. Black: CNTR-CMs; Red: K219T-CMs. **d** Conduction velocities recorded in cell strands consisting of CNTR- (black bar) and K219T- (red bar) CMs (95.9 ± 48.5 mm s$^{-1}$, $n = 47$ vs. 51.7 ± 34.7 mm s$^{-1}$, $n = 40$). All values are reported as mean ± SD. **$p <$ 0.005 (unpaired $t$-test). Data are relative to CMs differentiated from 2 independent lines from 2 CNTR and 3 K219T-LMNA subjects

constitutive (cLADs) and variable (vLADs) types, the latter corresponding to LAD borders; these have been shown to be enriched in H3K27me3 and to dynamically shift to and from the nuclear periphery during development, determining distinct gene expression outcomes[44,47]. Of note, impairment of this dynamic reorganization of the genome has been shown during terminal differentiation of skeletal myoblasts from Emery Dreifuss Muscular Dystrophy patients, in human adipose stem cells and during differentiation of iPSCs towards the endothelial lineage in the presence of the lypodystrophy-causing R482W *LMNA* mutation[43,48,49]. Here we found that, in CNTR-CMs, Lamin A/C interacted with the *SCN5A* locus at early stages of cardiac differentiation, but that binding was lost from day 12. In contrast, in K219T-CMs such binding was maintained throughout differentiation (Fig. 5c) and later in terminally differentiated CMs (Fig. 5a). The finding is coherent with the hypothesis that *SCN5A* may be included in a LAD that shifts from the nuclear periphery towards the nuclear interior during differentiation, but whole-genomic data from Lamin A/C ChIP-sequencing would be necessary to prove that *SCN5A* is truly located in a LAD domain. Nevertheless, this is consistent with ChIP-seq data on other cell types[18].

In addition, 3D-FISH (Fluorescence In-Situ Hybridization)[47,50] mapping of the position of *SCN5A* within the nucleus revealed a preferential positioning of the locus at the nuclear periphery in K219T-CMs vs CNTR-CMs (Fig. 5d).

To provide evidence on the role played by Lamin A/C in vivo, we used immunofluorescence microscopy and pathology tissue-chromatin immunoprecipitation (PAT-ChIP) to assess *SCN5A* gene regulation on paraffin-embedded sections of heart tissue obtained from the patients carrying the K219T Lamin A/C mutation. Coherently with our in vitro evidence, there was

reduced expression of Na$_v$1.5 in adult CMs from K219T-LMNA patients (Fig. 6a, b). This was associated with an increased binding of Lamin A/C at the *SCN5A* promoter region (Fig. 6c). These findings further support the hypothesis that K219T-mutated Lamin A/C more strongly associates to the *SCN5A* promoter, causing repression of its transcription.

**K219T-Lamin A/C cooperates with PRC2 in repressing *SCN5A*.** To deepen understanding of the molecular mechanism by which K219T-Lamin A/C affects transcriptional regulation of *SCN5A*, we next determined if the gene's expression was modulated by cross-talk between Polycomb Group (PcG) proteins and Lamin A/C. PcG proteins are epigenetic repressors that control the expression of a large number of genes during development, differentiation and disease. Interaction of PcG proteins with Lamin A/C is required for correct compartmentalization and assembly of the former[40,51]. Indeed, deficiency in Lamin A/C leads to disassembly and dispersion of PcG proteins and is associated with an impaired myogenic transcriptional program[40,51].

Here, we focused on Polycomb Repressive Complex 2 (PRC2), since it specifically catalyzes the methylation of lysine 27 on histone 3 (H3K27me3), the repressive mark we found enriched at the *SCN5A* promoter in K219T-CMs (see Fig. 5b). We thus evaluated the binding of PcG proteins to the *SCN5A* promoter by ChIP for Suz12, a component of PRC2. We found increased binding of Suz12 in K219T-CMs (Fig. 7a), further supporting the notion of cooperation between Lamin A/C and PcG proteins in modulating transcriptional regulation in CMs.

In addition, immunoprecipitation (IP) experiments confirmed this view and revealed that K219T-Lamin A/C is more bound to *Ezh2* (Enhancer of zeste homolog 2, the catalytic subunit of PRC2) than is wild-type Lamin A/C (Fig. 7b). This finding

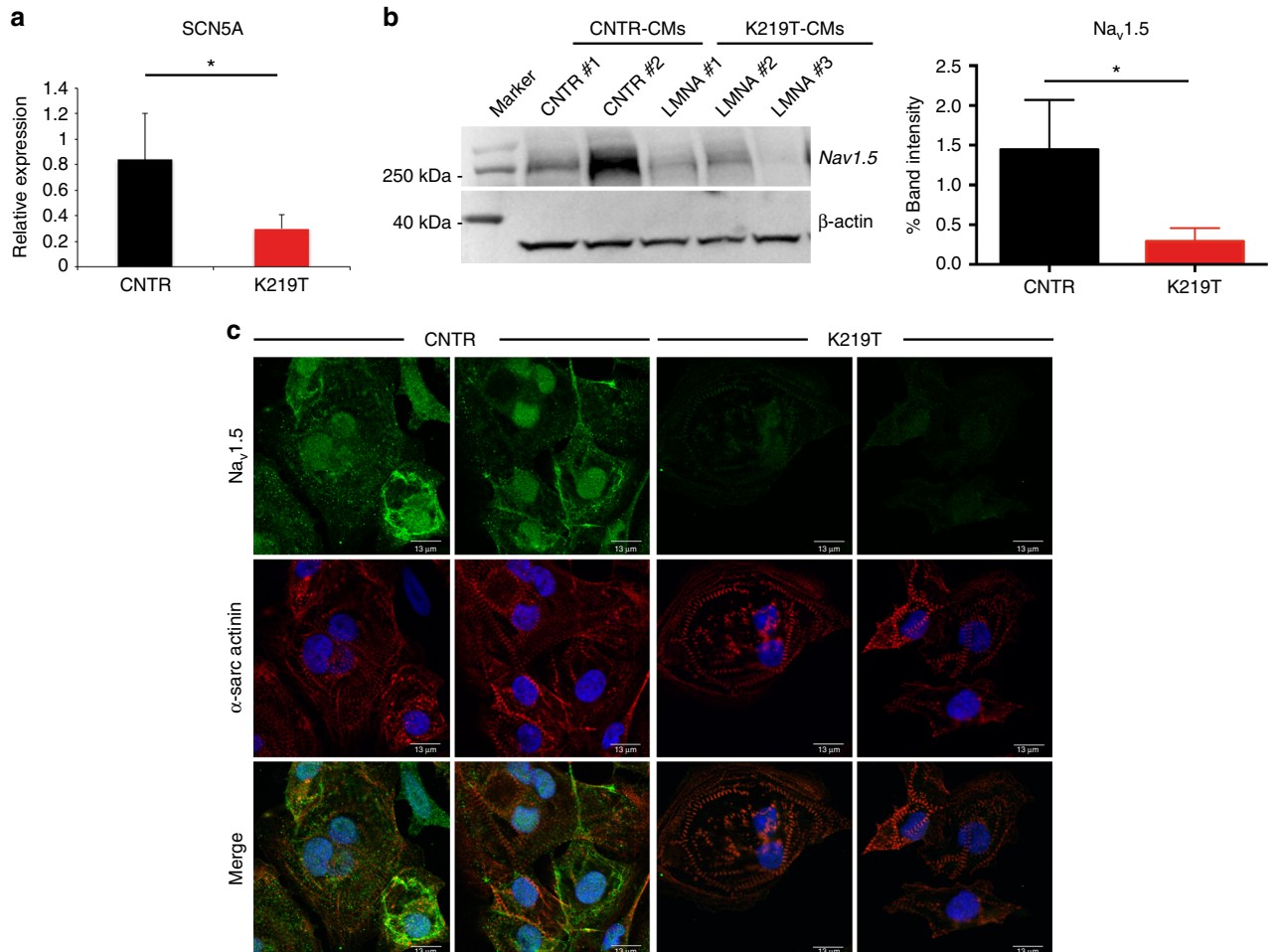

**Fig. 4** Reduced sodium current density is dependent upon reduction of *SCN5A* expression. **a** RT-qPCR showing reduced expression of *SCN5A* gene in CMs generated from K219T-LMNA-iPSCs vs those differentiated from CNTR lines. Data are represented relative to CNTR-CMs and normalized for expression of the housekeeping genes *18S* and *HGPRT*. Data are relative to CMs differentiated from 2 independent lines from 2 CNTR and 3 K219T-LMNA subjects. **b** Left: Representative Western blot showing reduced Na$_v$1.5 channel expression in K219T-CMs. Data are representative of one cell line per subject. Right: Na$_v$1.5 expression quantification in CNTR-CMs and K219T-CMs, calculated as densitometric Na$_v$1.5 /β-actin ratio (the diagram represents the mean of three independent experiments). **c** Representative immunofluorescence images for Na$_v$1.5 (green) and α-sarcomeric actinin (red) in CNTR-CMs and K219T-CMs; nuclei are stained with DAPI (Scale bars: 13 μm). A decrease in the expression of Na$_v$1.5 in K219T-CMs vs. CNTR-CMs is evident. All values are reported as mean ± SD. *$p < 0.05$ (unpaired *t*-test)

supports the hypothesis that the two proteins interact with higher affinity in the presence of K219T.

The observation was corroborated by super-resolution microscopy experiments, which revealed more-abundant co-localization of the Lamin A/C and Suz12 in K219T-CMs vs. CNTR-CMs (Fig. 7c). Of note, PRC2, alone or bound to Lamin A/C, was differentially distributed in CNTR-CMs and K219T-CMs, indicative of different protein dynamics in the two conditions (Fig. 7d, Supplementary Fig. 11). In detail, a proportion of the PRC2–LaminA/C complex was in closer proximity to the nuclear periphery in K219T-CMs vs. CNTR-CMs (Fig. 7d), a finding fitting well with results from 3D-FISH experiments, which locate *SCN5A* preferentially at the nuclear periphery in K219T-CMs, and sustains the notion that PRC2 and Lamin A/C cooperate to regulate *SCN5A* transcription by simultaneously binding to its promoter at the nuclear periphery. Moreover, a subpopulation of the complex significantly relocated to the nuclear interior in K219T-CMs (Fig. 7d, Supplementary Fig. 11), further supporting the notion that global rearrangement of the two proteins is likely to alter global transcription in the presence of *LMNA* mutations.

**K219T-Lamin A/C transduction mimics the K219T-CM phenotype**. To further prove that mutant Lamin A/C exerts a repressive activity on $I_{Na}$ density by downregulating *SCN5A* expression, we overexpressed K219T-LMNA in CMs derived from the RUES2 human embryonic stem cell line and assessed phenotype at the functional and molecular levels. To this end, we generated a Lamin A/C-EGFP (enhanced green fluorescent protein) C-terminal fusion lentiviral construct carrying the K219T mutation (Fig. 8a), and transduced RUES2-derived CMs. Transduction was carried out at day 5–6 of cardiac differentiation induction, a time-point at which Lamin A/C starts to become expressed in these cells (Supplementary Fig. 12A). When we compared Na$_v$1.5 current density in GFP$^{pos}$ (expressing the mutation) and GFP$^{neg}$ (wild-type) CMs, we found significantly decreased $I_{Na}$ in the former (at −30mV holding potential, 29.8% decrease vs GFP$^{neg}$; Fig. 8b, c), with no changes of the voltage dependence of activation and inactivation parameters (half-maximal voltage [$V_{1/2}$] and slope factor [$k$]) between the two conditions (Fig. 8d, Supplementary Table 4). Coherently, expression of *SCN5A* was significantly lower in GFP$^{pos}$ CMs (Fig. 8e).

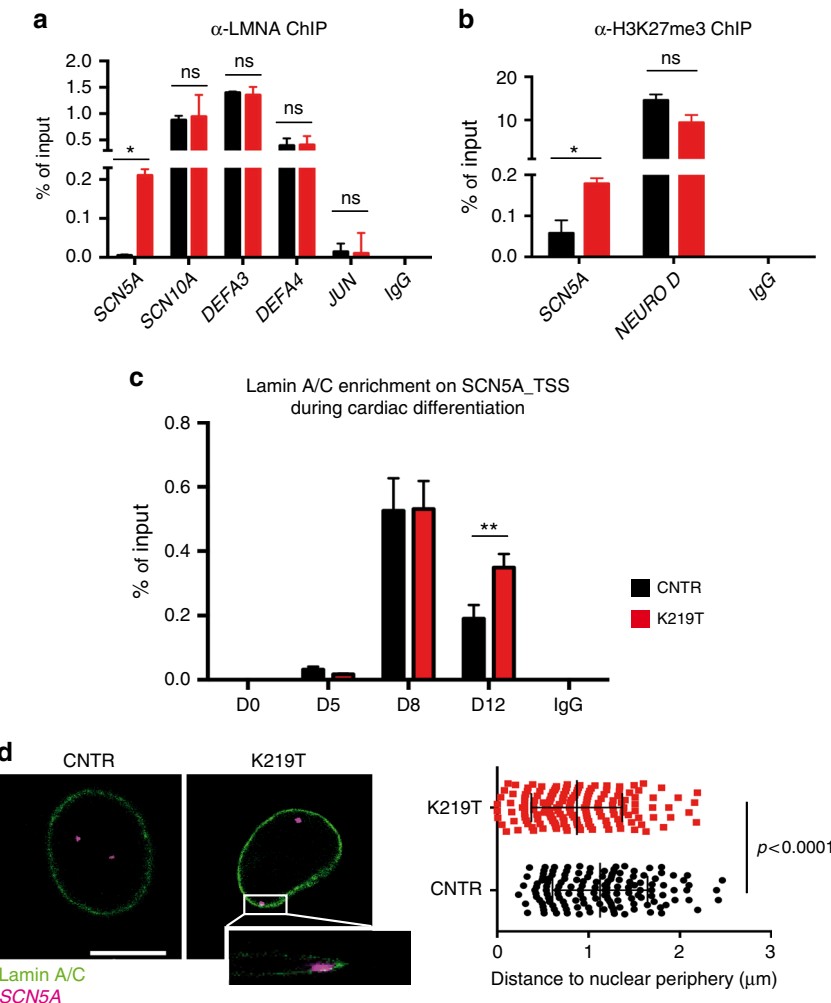

**Fig. 5** Epigenetic modulation of *SCN5A* gene expression. **a**, **b** ChIP analyses in CNTR-CMs and K219T-CMs with antibodies against Lamin A/C (**a**) and H3K27me3 (**b**) targeting the *SCN5A* genomic region. Data are presented as a percentage of input chromatin precipitated and relative to 2 independent iPSC clones from each subject. ChIP for Lamin A/C, indicating increased binding of Lamin A/C to the promoter region of the *SCN5A* gene in K219T-CMs; no enrichment was detectable for *JUN*, used as a negative control region for the ChIP experiment; *SCN10A*, *DEFA3*, and *DEFA4* were used as positive control regions for the ChIP experiments, and they show comparable enrichment of Lamin A/C in CNTR-CMs and K219T-CMs (**a**). ChIP for the histone modification H3K27me3, showing an enrichment of the repressive histone mark at the *SCN5A* gene promoter in K219T-CMs, supporting the reduced expression of the gene detected in these cells; *NEURO D* was used as positive control region for the antibody (**b**). **c** ChIP against Lamin A/C, showing the dynamics of Lamin A/C enrichment at the transcription start site (TSS) of *SCN5A* during cardiac differentiation (from days 0 to 12). The graph is relative to one representative experiment (out of three). For this analysis, two-way ANOVA was used for the statistic test. **p < 0.01. **d** 3D-FISH for Lamin A/C (green) and *SCN5A* (magenta) conducted with an antibody and a DNA probe, respectively. Left: representative images of nuclei from CNTR and K219T-CM preparations (Scale bar: 9 μm). Magnified box shows the colocalization of the *SCN5A* probe at the nuclear lamina. Right: Analysis of the distance from the nuclear lamina, showing preferential localization of the *SCN5A* probe (magenta) at the nuclear periphery in K219T-CMs vs. CNTR. Quantification was carried out using a specific plug-in of the ImageJ software (CNTR-CMs: $n = 72$ cells; K219T-CMs: $n = 82$ cells; $p < 0.0001$). All values are reported as means ± SD, if not differently specified. *$p < 0.05$; **$p < 0.01$, ns, not significant (unpaired *t*-test)

By contrast, overexpression of either an empty GFP vector or a vector encoding wild-type LMNA did not produce any effect on $Na_v1.5$ current density in RUES2-CMs (Supplementary Fig. 12, Supplementary Table 4), further indicating that the observed phenotype is causally linked to K219T on *LMNA*.

**SCN5A overexpression rescues the functionality of LMNA-CMs**. To corroborate the hypothesis that the effect on $I_{Na}$ is directly mediated by downregulation of *SCN5A*, we overexpressed C-terminus GFP-tagged wild-type *SCN5A* in LMNA-CMs carrying either K219T or R190W (Supplementary Fig. 13A), and measured $I_{Na}$ density by patch-clamp. We found that overexpression of *SCN5A* in the presence of either of the two mutations was sufficient to significantly increase $I_{Na}$ density to a level

that was even higher than that in controls (Supplementary Fig. 13B–D, Supplementary Table 4). Of note, $Na_v1.5$-dependent AP properties were also restored by *SCN5A* overexpression (Supplementary Fig. 13E, Supplementary Table 5).

**K219T correction rescues functional and molecular defects**. To unequivocally establish the direct link between the mutation and the phenotype described above, we generated isogenic lines in which the point mutation was specifically corrected (here referred to as K219T-corrected), using a CRISPR-Cas9-mediated strategy (Fig. 9a, b and Supplementary Fig. 14)[52,53].

Comparative functional and molecular analyses of CMs differentiated from K219T-corrected LMNA-iPSCs demonstrated that correction of the mutation is sufficient to restore $I_{Na}$ density

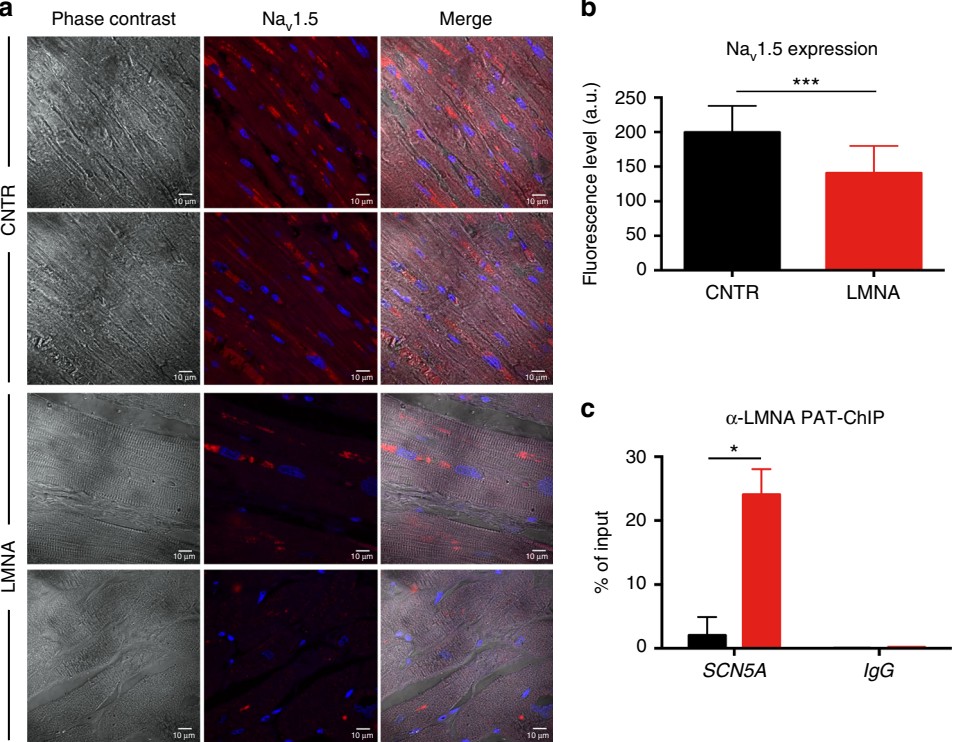

**Fig. 6** Lamin A/C-mediated reduction of $Na_v1.5$ expression is confirmed in heart sections. **a** Immunostaining of paraffin-embedded heart sections from patients carrying the K219T mutation on Lamin A/C (bottom panels) and from two control subjects (top panels) targeting the $Na_v1.5$ sodium channel (red); nuclei were stained with DAPI (scale bars: 10 μm). **b** Histogram showing the quantification of the red fluorescent signal shown in **a**, corresponding to the $Na_v1.5$ protein, confirming reduced expression also in K219T-LMNA patient hearts. (a.u. is for arbitrary units). Data are relative to heart sections from three healthy controls ($n = 11$) and four mutant samples ($n = 12$). **c** PAT-ChIP analysis of heart sections from patients carrying the K219T mutation on *LMNA* (red bar) and from two control subjects (black bar), showing increased binding of Lamin A/C at the *SCN5A* promoter region in K219T-CMs. Chromatin was immunoprecipitated with an antibody against Lamin A/C and the resulting DNA was analysed by realtime PCR and presented as a percentage of input chromatin precipitated for the indicated region. All values are reported as means ± SD. *$p < 0.05$; ***$p < 0.0001$ (unpaired *t*-test)

and *SCN5A* expression to levels indistinguishable from those in CNTR-CMs (Fig. 9c, d and Supplementary Table 4). Of note, restoration of functional and molecular phenotypes was coupled with decreased binding of Lamin A/C to the *SCN5A* promoter (Fig. 9e), strongly supporting the hypothesis that dysregulation of *SCN5A* and the consequential effect on $I_{Na}$ is caused by the mutation.

Taken together, these findings support a mechanistic model in which mutated Lamin A/C perturbs *SCN5A* expression by favouring the binding of PRC2 to its promoter, leading to decreased peak $I_{Na}$ and slower conduction velocity (Fig. 10).

## Discussion

Understanding the molecular mechanisms underlying the electrophysiological defects in LMNA-CMP is essential for improving current therapeutic decisions. Patients who develop left ventricular dysfunctions are currently treated with the standard medical therapy for chronic heart failure, typically β-blockers and inhibitors of the renin-angiotensin system, which are usually not sufficient to prevent heart transplantation. Moreover, most patients eventually require a pacemaker or implantable cardioverter defibrillator to prevent fatal arrhythmias.

For this study, we developed a patient-specific model of cardiac laminopathy to determine how the heterozygous K219T mutation on Lamin A/C contributes to the pathogenesis of the clinical phenotypes associated to LMNA-CMP. Through comprehensive electrophysiological characterization of mutant iPSC-derived CMs, we identified a series of consequential

functional defects affecting excitability and cardiac impulse propagation that were caused by Lamin A/C-driven repression of *SCN5A* expression, leading to reduced peak $Na_v1.5$ density. The major electrical phenotype that emerged consisted in a reduced $dV/dt_{max}$, which we found to be linked to a profound impairment of $I_{Na}$ density. Importantly, the decreased cellular excitability seen in LMNA-CMs is coherent with the different AP characteristics and the biophysical parameters of all analysed cell populations (i.e., spontaneously active, electrically adjusted and working-like iPSC-CMs), further validating the use of this cellular model to study $Na_v1.5$ current and conduction in LMNA-CMP.

While previous studies in other cellular systems have already linked Lamin A/C mutations to $Na_v1.5$ regulation, results have been contradictory. Indeed, Liu and colleagues reported that R644C and R190W exerted an inhibitory effect on $Na_v1.5$ when overexpressed in HEK293T cells[54]; a more recent study carried out on ventricular CMs from homozygous[N195K/N195K] mice produced opposite results, with increased peak $I_{Na}$ and prolongation of AP duration[55]. Our findings clarify these discrepancies and provide evidence for the phenotype in a relevant human pathophysiological context, namely CMs derived from an iPSC-based model of laminopathy.

It is well established that the density of voltage-gated $Na^+$ current and inactivation of $Na^+$ channels at depolarized MDPs are at the basis of CM excitability and are a major determinant of impulse propagation across the myocardium[38]. Many mutations on *SCN5A* affecting expression or function of the $Na_v1.5$ channel have been found in patients with various forms of inherited

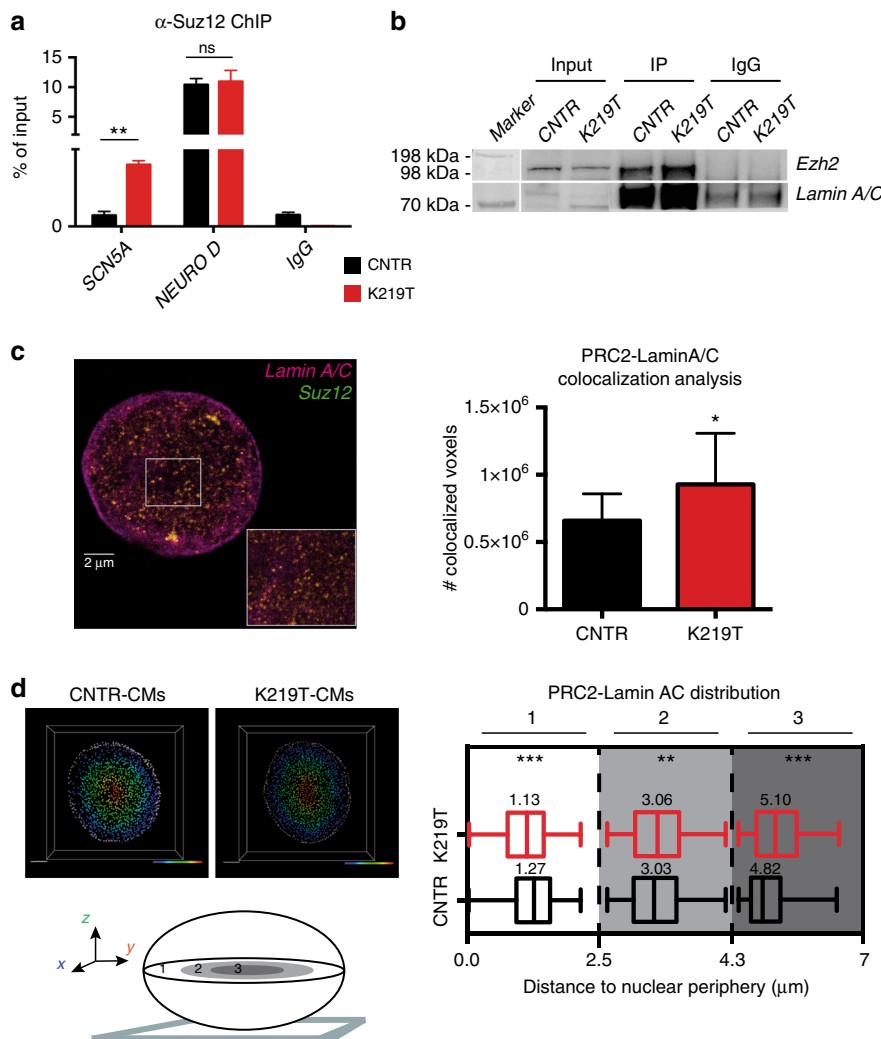

**Fig. 7** Lamin A/C–PRC2 interplay in the regulation of *SCN5A* transcription. **a** ChIP against Suz12, showing enrichment at the *SCN5A* promoter region in K219T-CMs. No significant differences were detectable at the *NEURO D* promoter, used as positive control for the antibody. Data are presented as percentage of input chromatin precipitated and relative to 2 independent iPSC lines per subject. (**\*\****p* < 0.01, unpaired *t*-test). **b** Co-immunoprecipitation (co-IP) of LaminA/C and Ezh2 (PRC2 subunit) in CNTR-CMs and K219T-CMs, showing a higher binding affinity of mutant LaminA/C for Ezh2. An unrelated IgG antibody was used as negative control. One representative (out of three) experiment is shown. **c** 3D-STED super-resolution microscopy for Lamin A/C and Suz12. Left: representative image of a nucleus stained for the two proteins. The inset shows co-localization of the two proteins (yellow spots). Right: bar graph indicating the quantification of voxels of colocalization and showing higher levels of interaction in K219T-CMs vs. CNTR-CMs. (*n* = 16 cells for each condition). (*\*p* < 0.05, unpaired *t*-test). **d** Distribution profile of the PRC2–Lamin A/C colocalization channel from 3D-STED microscopy acquisitions. For the analysis, the distance transformation function of Imaris software was applied to the z-stack images. Left: Strategy to determine the distance of the PRC2–Lamin A/C colocalization signal from the nuclear periphery. Top left: representative images of CNTR-CM and K219T-CM nuclei (scale bar 2 μm). Gray spots indicate Lamin A/C fluorescent signal: this was selected as the outside ring and served as the nuclear edge. The colored bar is indicative of the relative position of the spots to the nuclear edge: farthest positions in violet, closest in red. Bottom left: strategy of data representation: distances were scored into three zones: (1) periphery (from 0 to 2.15 μm, white); (2) middle (from 2.15 to 4.3 μm, light gray); and (3) central zone (from 4.3 to 6.45 μm, dark gray). The distance of every single colocalization spot was calculated from the nuclear edge. Right: box plot graphs representing PRC2–Lamin A/C distribution in the three selected zones. For each box plot, bottom and top of the boxes respectively indicate the lower and upper quartiles of the data set and whiskers represent the highest and the lowest values. Center line is the median value. Values are referred to median ± SD (*n* = 5 cells per condition, obtained from 5 independent iPSC clones). (*\*\*\*p* < 0.0001; *\*\*p* < 0.01, two-way ANOVA)

arrhythmias;[56] moreover, conduction defects and sudden death also represent phenotypic hallmarks of LMNA-CMP[4]. These notions strongly support a pathogenic role for downregulated Na⁺ channel expression in the onset of conduction disturbances seen in K219T-CMs.

Along this line, conduction velocities (CVs) were significantly decreased in confluent strands of K219T-CMs. As expected, most likely because of the fetal-like maturation stage of CMs differentiated from iPSCs[34], CVs in CNTR multicellular strands

(~100 mm/s) were found to be slower than the physiological CVs of neonatal CMs cultured on the same strands (~450 mm/s) or on anisotropic intact myocardial tissue (~600 mm/s); nonetheless, CVs recorded in our experiments were faster than those reported by others in multicellular preparations of stem cell-derived CMs (~50 mm/s)[33,57]. These differences are most likely due to different cellular excitabilities, the particular electrical coupling among the cells and the patterned geometry used for culturing confluent cells.

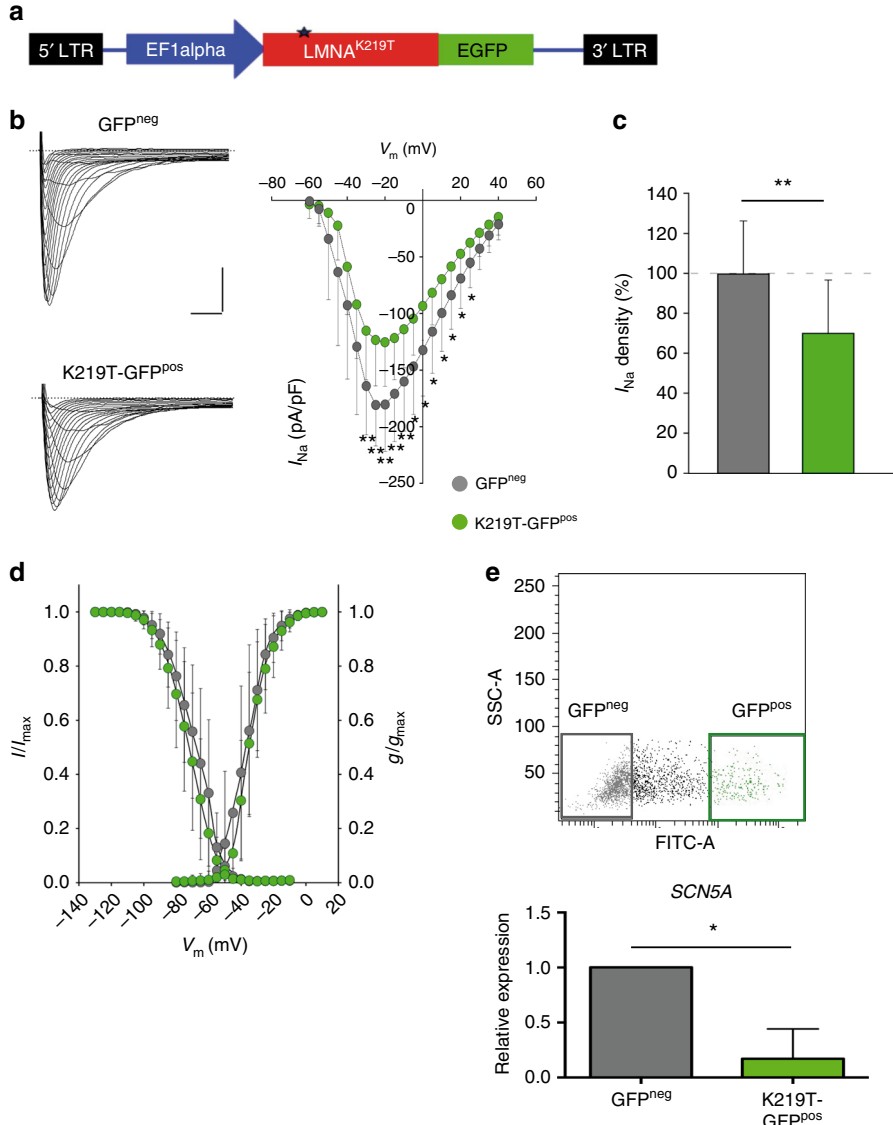

**Fig. 8** K219T-LMNA overexpression in RUES2-CMs reproduces phenotypes of K219T-CMs. **a** Lentiviral vector expressing the C-terminal GFP-tagged K219T-Lamin A/C used for lentiviral particle production. **b** Analysis of sodium current ($I_{Na}$) densities in CMs differentiated from the human embryonic stem cell line RUES2 and transduced with lentiviral particles expressing K219T-LaminA/C-GFP. Left column: examples of $I_{Na}$ traces recorded in GFP$^{neg}$ (top) and GFP$^{pos}$ CMs (bottom) (Scale bar, 2 ms, 50 pA/pF). Right panel: I-V curves constructed from average peak sodium current density as a function of voltage command measured in GFP$^{neg}$ and GFP$^{pos}$ CMs. (GFP$^{neg}$: $n = 14$; GFP$^{pos}$: $n = 21$), showing significant Na$^+$ current reductions in GFP$^{pos}$ cells. **c** $I_{Na}$ density, measured at $-30$ mV, in GFP$^{pos}$ ($115.20 \pm 43.35$ pA/pF) vs. GFP$^{neg}$ ($164.02 \pm 43.18$ pA/pF) CMs, expressed as a percentage. **d** Voltage dependences: steady state activation (GFP$^{neg}$: $n = 14$; GFP$^{pos}$: $n = 21$)/inactivation (GFP$^{neg}$: $n = 11$; GFP$^{pos}$: $n = 17$) curves. **e** RT-qPCR comparing *SCN5A* expression in GFP$^{neg}$ and GFP$^{pos}$ CMs isolated through FACS sorting. Data are represented relative to GFP$^{neg}$ samples and normalized to expression of *HGPRT* and *18S* housekeeping genes. Top: gating strategy and cell populations selected for the sorting are shown (sequential sorting strategy is provided in the Supplementary Fig. 15). All values are reported as mean ± SD. *$p < 0.05$; **$p < 0.005$ (unpaired *t*-test)

Impulse propagation across the myocardium depends not only on CM excitability but is also determined by intercellular conductance[38,58]; previous studies have linked *LMNA*-dependent conduction defects to misexpression or mislocalization of Connexin-43, a gap junctional protein that, in the heart, is localized at the intercalated disks (end-to-end coupling) of CMs and is critical for establishing intercellular conductance and driving impulse propagation[14,38,58,59]. Even though we did not observe altered Connexin-43 expression in our model, we cannot exclude the possibility that reduced gap junctional coupling due to mislocalization or internalization of the connexons may be involved in determining the observed conduction defects. In support of this assumption, we found that the slowing of CVs in K219T-

CMs was more profound than the reductions in d$V$/d$t_{max}$ and $I_{Na}$ density in single-cell preparations. The stronger effect on CV than on excitability parameters may be supported by an additive effect of the two impulse propagation pathways (excitability and intercellular conductance).

At the molecular level, we found an epigenetically-mediated regulatory effect on *SCN5A* that affected the expression of the Na$_v$1.5 channel, leading to diminished $I_{Na}$ density and impaired impulse propagation. Besides a role in maintaining the structural stability of the nucleus, Lamin A/C may have key regulatory roles on gene transcription, either by direct interaction with DNA and chromatin or through the binding to chromatin complexes or transcription factors[40,47,50,60,61]. Genome–Lamin A/C interaction

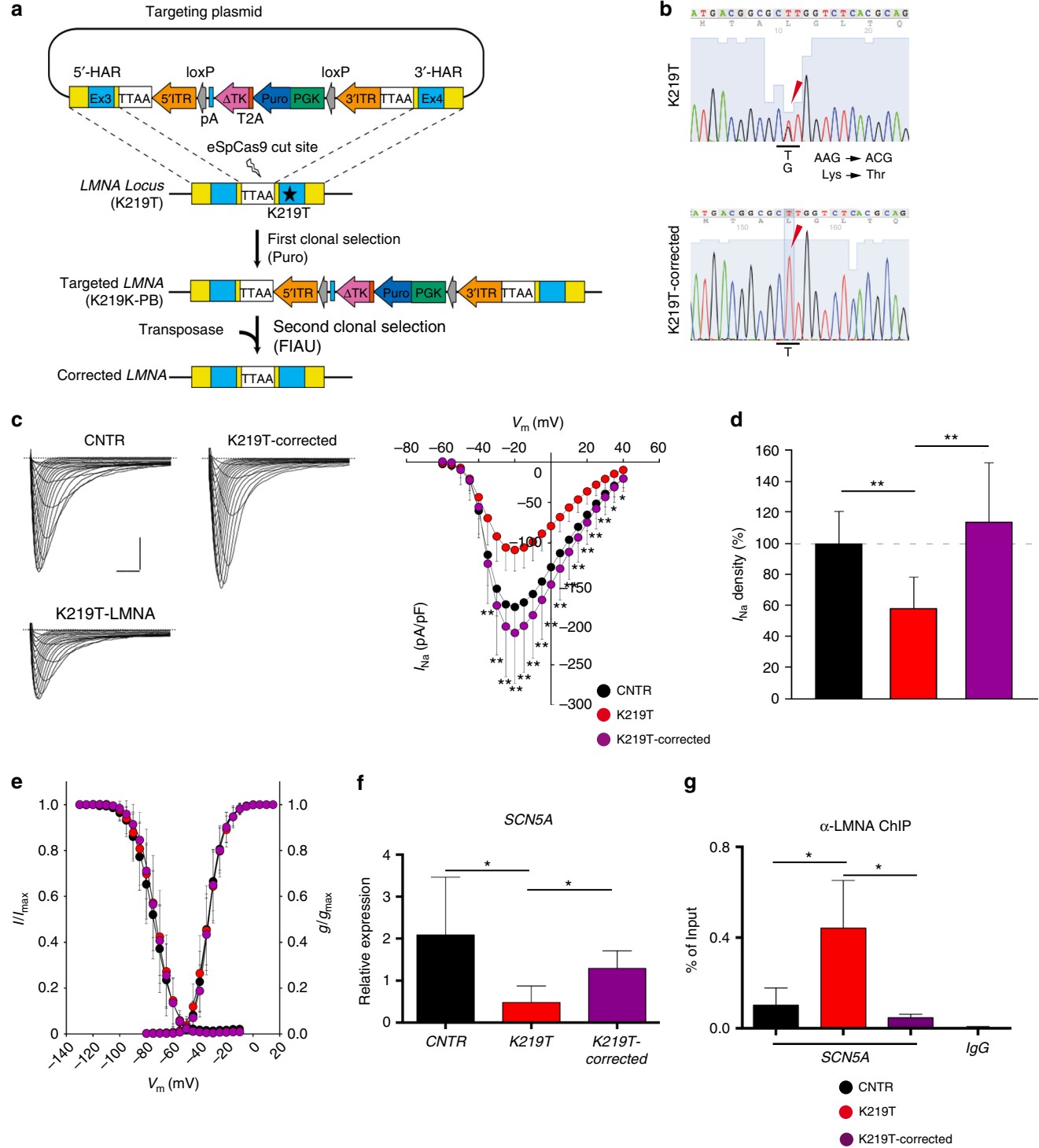

occurs through large chromatin domains and in spatially restricted regions on gene promoters, modulating gene expression[18,41]. *LMNA* missense mutants have been found aberrantly associated with specific portions of the genome, leading to transcriptional defects, rearrangement of LADs and Lamin–chromatin associations occurring at the nuclear interior[43,62]. It was therefore suggested that changes in gene expression observed in cells from patients carrying different types of Lamin A/C mutation (i.e., progeroid disorder, lipodystrophies, and muscle dystrophy) were linked to gain/loss of LADs, affecting genome architecture[43,48,49,62,63]. Even though genome-wide investigations are necessary to identify lamin associated domains and firmly establish their remodeling, our findings

well suit this regulatory circuit, providing a new piece of evidence for Lamin A/C-dependent modulation of the expression of a key gene in cardiac electrophysiology, *SCN5A*, as a determinant of the disease.

In the mechanism we propose here, mutant Lamin A/C proteins form a repressive complex with PRC2, which binds to the *SCN5A* promoter and catalyzes the deposition of H3K27me3. In this conformation, *SCN5A* is preferentially located at the nuclear periphery. This further contributes to a repressive environment, resulting in the downregulation of targeted-gene transcription and leading to the diminished $I_{Na}$ and impaired propagation registered in *LMNA*-mutant cells.

**Fig. 9** Correction of K219T mutation rescues Na$^+$ current phenotype and *SCN5A* expression. **a** Gene editing strategy used for the correction of the mutation. **b** Representative electropherograms from DNA sequencing of exon 4 of the *LMNA* gene showing the correction of the alanine-to-cytosine transversion at position 656 (c.656A<C) in the generated isogenic K219T-corrected iPSC lines (the reverse sequence is shown in the figure; the T<C mutation is indicated by the arrow). The nucleotide mutation corresponds to the amino acid change p.K219T. The electropherogram from the parental K219T-LMNA iPSC line is shown on the top panel as reference. **c** Left panel: examples of Na$^+$ current ($I_{Na}$) traces recorded in CNTR- (top), K219T-corrected (top) and K219T- (bottom) CMs (Scale bar, 2 ms, 50 pA/pF). Right panel: *I-V* curves constructed from average peak sodium current density as a function of voltage command measured in the three conditions described above (CNTR: $n = 23$; K219T-corrected: $n = 14$; K219T: $n = 22$), showing significant sodium current increases in K219T-corrected cells with respect to the parental mutant CMs. *adj.$p < 0.05$; **adj.$p < 0.005$ (two-way ANOVA). **d** $I_{Na}$ density, measured at −30 mV, in K219T-corrected CMs (176.94 ± 61.05 pA/pF) relative to K219T-CMs (91.58 ± 30.69 pA/pF) and CNTR-CMs (156.12 ± 33.73 pA/pF) expressed as a percentage. **e** Voltage dependences: steady state activation (CNTR: $n = 23$; K219T-corrected: $n = 14$; K219T: $n = 22$)/inactivation (CNTR: $n = 18$; K219T-corrected: $n = 12$; K219T: $n = 17$) curves. **f** RT-qPCR showing that *SCN5A* gene expression is restored to control levels in CMs generated from K219T-corrected-iPSCs. Data are represented relative to CNTR-CMs and normalized to expression of the housekeeping genes 18S and HGPRT. **g** ChIP against Lamin A/C, showing loss of Lamin A/C binding at the *SCN5A* promoter region in K219T-corrected CMs vs mutant K219T-CMs. Levels of Lamin A/C binding are instead indistinguishable from those detected in CMs differentiated from control cells. All values are reported as means ± SD. *adj.$p < 0.05$; **adj.$p < 0.005$ (one-way ANOVA), unless otherwise specified

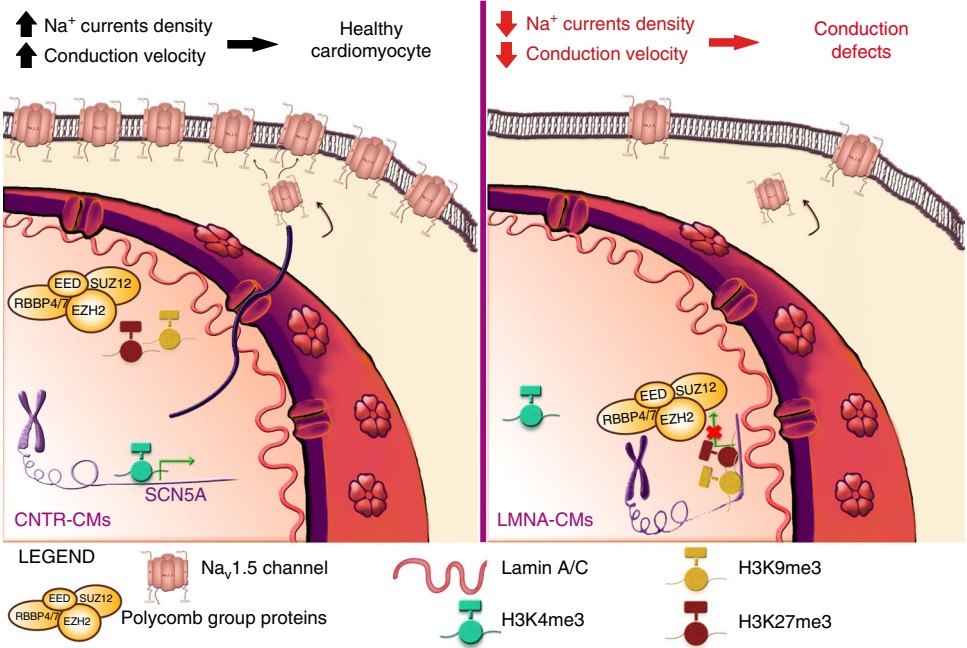

**Fig. 10** Proposed pathogenic mechanism of action of K219T-Lamin A/C in CMs. Left: healthy CMs. The *SCN5A* genomic region is localized in the nuclear interior and actively transcribed; the Na$_v$1.5 sodium channel density at the plasma membrane is sufficient to assure proper impulse propagation. Right: Laminopathic CMs. Mutant Lamin A/C and PRC2 bind each other with high affinity and are enriched at the *SCN5A* promoter region, where the H3K27me3 (catalyzed by PRC2) and H3K9me3 repressive histone marks are also present. In this repressive environment, this genomic region is preferentially sequestered at the nuclear periphery. As a result, *SCN5A* gene transcription is repressed, leading to reduced expression of the Na$_v$1.5 sodium channel at the plasma membrane, which, in turn, has an effect on reducing conduction velocity. This regulatory mechanism may be at the basis of the onset of the conduction defects in patients with LMNA-CMP

ChIP and 3D-FISH experiments demonstrated selective and dynamic binding of Lamin A/C to the *SCN5A* promoter in CMs and throughout cardiac differentiation. However, we cannot exclude that additional layers of chromatin regulation may be involved. Indeed, since the nuclear lamina operates as a docking platform for the binding of transcription factors and proteins involved in transcriptional regulation, it is possible that other factors (e.g., lamin-binding proteins, other chromatin regulators, lncRNAs) may further regulate *SCN5A* expression.

Results from super-resolution STED microscopy further support the view that a regulatory circuit involving Lamin A/C and PRC2 is acting on *SCN5A* at the nuclear periphery, indicating a more abundant peripheral distribution of PRC2, by itself or bound to Lamin A/C, in K219T-CMs. More generally, the distribution profiles of the PRC2–Lamin A/C complex indicated that dynamics significantly differ in wild-type and K219T CMs and distinguish

two conditions, one in which PRC2–Lamin A/C is closer to the nuclear periphery and another in which there is a tendency to relocate towards the interior. These findings are consistent with the previously described link between Lamin A/C and the intra-nuclear localization of PcG proteins and with their nuclear compartmentalization reported in cells lacking Lamin A/C[51]. The concept that PRC2 and Lamin A/C may interact also at the nuclear periphery is instead novel; although further investigations are required, it is possible that the higher affinity of Lamin A/C for PRC2 in the presence of the mutation results in persistent binding of the two proteins together regardless of nuclear localization, and that, in this form, they relocate through the nucleus to exert their repressive effect. In addition, results from 3D-STED are also indicative of different dynamics for mutant Lamin A/C and the wild-type protein; this is in agreement with a recent study describing the different dynamics of diverse mutated LMNAs[64].

As a result of this global rearrangement, an effect on other disease-relevant genes cannot be ruled out: transcriptional reprogramming is likely to occur at a global level in differentiating cardiac progenitors and CMs in the presence of mutated Lamin A/C, leading to misregulation of genes involved in other disease-relevant processes, such as contractility and cell metabolism.

Besides this speculative view, our findings improve the understanding of the pathogenesis of LMNA-CMP-related conduction defects and indicate a new pharmacological target. Indeed, it is conceivable that clinically relevant results could be had by either selectively activating the Na$_v$1.5 channel through specific modulators or inducing *SCN5A* expression (e.g., with epigenetic drugs or nucleic acid mimics). Normalization of the electrophysiological phenotype of mutated CMs through over-expression of *SCN5A* together with the evidence that this gene is positively modulated in isogenic K219T-corrected CMs point to *SCN5A* as a potential gene-therapy target for LMNA-CMP.

In conclusion, we describe a new pathogenic mechanism for the conduction defects associated with LMNA-CMPs, dependent upon transcriptional changes driven by mutant Lamin A/C. Whether the effect on the *SCN5A* is selective for certain mutations affecting specific protein domains or is more broadly linked to altered Lamin A/C conformation is still unclear and needs further investigation.

## Methods

**iPSC lines generation and maintenance**. iPSCs were generated from skin fibroblasts of three LMNA patients carrying the K219T mutation and two healthy family members (without the mutation) using the STEMCCA polycistronic lentiviral vector[65]. Reprogrammed cells were selected by their morphology and tested for their actual pluripotency[21,66]. Details on generation, characterization and maintenance of the lines are provided in the Supplementary Extended Methods.

**Generation of isogenic control iPSC lines**. The heterozygous K129T mutation in the *LMNA* gene was corrected back to the wild-type allele using a CRISPR/Cas9-facilitated gene editing strategy based on a previously described method[53,67,68].

The results described were obtained from line LMNA#1_2. Briefly, we developed a two-step approach to change a single nucleotide in the absence of any other genetic "scarring" (Fig. 8a and Supplementary Fig. 14A). First, we corrected the mutation by homozygous insertion of a plasmid-based targeting cassette carrying the wild-type sequence. This was obtained via homologous directed repair followed by positive selection of gene-targeted clones based on a puromycin resistance gene carried within by a piggyBac element. Secondly, we excised such piggyBac cassette using transposase, leaving behind only a "TTAA" remnant sequence. For this, corrected clones were enriched by a negative drug selection strategy based on the human herpes simplex virus thymidine kinase gene also carried by the piggBac. After each gene-targeting step, multiple clonal lines were isolated and genotyped using a combination of genomic PCR assays. Uncropped gels are provided in the Source Data file. Positive clones were further characterized by targeted Sanger sequencing and karyotyped to confirm their genomic stability.

A more detailed description of this procedure is provided in the Extended Methods of the Supplementary Information.

**Cardiac differentiation**. Differentiation into CMs was achieved using a chemically-defined serum free protocol, which is based on Wnt pathway modulation in RPMI medium[27,66]. In brief, 95–100% confluent iPSCs were induced with the Wnt activator CHIR99021 for 24 h and then, at day 3, exposed to IWR-1, which inhibits the Wnt pathway, for 48 h. At day 10 of induction, medium is supplemented with insulin. Spontaneous contracting activity usually appears around day 8–10 of the differentiation process. Efficiency of the differentiation was determined for all lines through the detection of the cardiac-specific marker α-sarcomeric actinin by FACS. CMs were used for the experiments 20–25 days after contraction started. Data presented are average of values obtained in CMs from 2 CNTR and 3 LMNA subjects; two iPSC lines per subject were used.

**Electrophysiological analyses**. Maximal diastolic potential (MDP), action potential (AP) properties, and whole-cell Na$^+$ current ($I_{Na}$) recordings were performed in low-density cell cultures using a Multiclamp 700B patch-clamp amplifier (Molecular Devices) and analysed with dedicated software (ClampFit 9.2). Whole cell currents were normalized to cell capacitance and are reported as pA/pF. Single iPSC-CMs were identified on the basis of their typical morphology[21]. Spontaneously active iPSC-CMs and iPSC-CMs in which the MDP was adjusted to

physiological resting cardiac membrane potentials by a constant hyperpolarizing current injection were selected for measurement of AP characteristics.

Impulse propagation characteristics along 80-μm wide strand preparations were assessed optically by a fast resolution camera (Ultima, Scimedia, USA). Preparations were surperfused at 36 °C using the voltage-sensitive dye di-8-ANEPPS. Conduction velocity was computed by linear regression of activation times recorded along the preparation. Optical action potential propagation derived from two strands was acquired for each field of view at 1 KHz sample resolution for 4 s recording. Magnification 10×, n.a. 0.5. $n = 47$ for CTRL-CMs, $n = 40$ for K219T-CMs.

In all electrophysiological experiments, preparations were superfused at 1.5 ml min$^{-1}$ with HBSS containing 10 mmol l$^{-1}$ of HEPES (pH 7.40) (Sigma Aldrich) (pH 7.40). For $I_{Na}$ recordings, a low Na$^+$ solution was used that contained (in mmol l$^{-1}$): NaCl 70, CaCl$_2$ 1.8, CsCl 90, MgCl$_2$ 1.2, glucose 10, HEPES 10, and nifedipine 0.001 (pH 7.4 with CsOH).

**Protein analyses**. Protein expression was analysed by Western blotting and immunofluorescence. For the Western blotting, 30 μg of lysate was used. Primary antibodies (Na$_v$1.5 1:1000, anti-rabbit Cell Signaling D9J7S, β-actin 1:2000, anti-goat Santa Cruz sc1615) were incubated overnight. For detection, HRP-conjugated secondary antibodies and the Millipore Chemiluminescent HRP substrate kit were used. Co-IP experiments were performed as previously[51]. Detailed protocol is available in the Supplementary Methods. Uncropped blots from Western Blot experiments are provided in the Source Data File.

For immunofluorescence, iPSC-CMs were fixed in 4% paraformaldehyde, blocked with 3% serum and permeabilized with 0.1% Triton. Stainings were performed using the following antibodies: anti-Na$_v$1.5 channel protein (Alomone #ASC005), anti-α-sarcomeric actinin protein (Abcam EA-53- ab9465), anti-Lamin A/C (Santa Cruz Biotechnologies, N18 sc6215). For detection, we used Alexa-Fluor 488-conjugated and 555-conjugated secondary antibodies (Molecular Probes from Thermo Scientific) raised in the appropriate species for the experiments. Images were acquired on an FV1000 confocal laser-scanning microscope (Olympus) and analysed using ImageJ software.

Binding of Lamin A/C and Suz12 was assessed by STimulated Emission Depletion (STED) super-resolution microscopy. Cells were incubated overnight at +4 °C with the primary antibodies against Lamin A/C (anti-mouse, from Santa Cruz 7292-X, 1:250) and Suz12 (anti-rabbit, from Cell Signaling #3737, 1:250). For detection, we used Abberior STAR RED-conjugated and STAR 580-conjugated secondary antibodies. Images were acquired on a Leica TCS SP8 STED 3× microscope and analysed using Imaris software. PRC2-Lamin A/C distribution profiles ware determined using the "Distance transformation" Xtension of Imaris Bitplane. Details of all experimental procedures and analyses are available in the Extended Methods of the Supplementary Information.

**Gene expression**. Gene expression studies were performed on cDNA synthesized from total RNA using SuperScript Vilo cDNA synthesis kit (ThermoFischer Scientific). Comparative gene expression was determined by realtime PCR using Sybr Green (ThermoFisher Scientific) and specific primers. A list of the primers used for the experiments is provided in the Supplementary Information files (Supplementary Table 6). The relative expression was calculated as 2$^{-\Delta\Delta Ct}$ using the comparative Ct method and 18S or HGPRT as housekeeping control as detailed in the respective figure legends.

**Chromatin Immunoprecipitation**. Two wells (of a 12-well plate) of confluent CMs were fixed in 1% formaldehyde for 10 min and then quenched with 0.125 M of glycine for 5 min. Chromatin was sonicated with Bioruptor Diagenode Sonicator and immunoprecipitated overnight at 4 °C with 4 μg of Lamin A/C antibody (anti Lamin A/C (636), Santa Cruz sc-7292×) or 5 μg of antibodies against the analysed histone modifications (anti-H3K4me3, Active Motif 39159; anti-H3K27me3, Abcam ab6002; anti-H3K9me3, Abcam ab8898; anti-Suz12, Cell Signaling, #3737).

The pathology tissue chromatin-immunoprecipitation (PAT-ChIP) procedure was carried out as previously described[69]. In brief, paraffin was removed with fresh Histolemon solution (Carlo Erba), chromatin extracted, sonicated and immunoprecipitated. For the immunoprecipitation step 3 μg of ChIP-grade antibody was added to 0.5–2.5 μg of tissue chromatin.

Quantitative PCR was performed in triplicate using SYBR Select Master Mix (ThermoFisher Scientific). Ct values were calculated using Viia$^{TM}$ 7 software and relative enrichment was calculated as ChIP/input ratio. Primer sequences are listed in the Supplementary Table 6.

**3D-FISH**. In-situ hybridization was essentially as previously described, with minor modifications[70,71]. Slides were scored under an Olympus FV-1000 confocal laser-scanning microscope with a ×100 oil immersion objective using a ×2 optical zoom with a z-step of 0.3 μm between optical slices as previously described[47].

**Statistics**. Statistical significance was determined with GraphPad Prism 6.0 software using unpaired two-tailed *t*-test (homoscedastic or heteroscedastic where appropriate) for the analysis of two groups or ANOVA for experiments with two between-subjects factors (two-way). Bonferroni's highly significant difference

(HSD) post hoc analysis was used for multiple pairwise comparison. Data are presented as mean ± standard deviation (SD) of at least 3 independent experiments, unless otherwise noted. Differences between data sets were considered significant at $p < 0.05$. NS represents no significant difference. Statistical parameters are reported in the respective figures and figure legends.

For electrophysiological analyses, the number of samples refers to independent experiments. Assessment of normality of the data was calculated with Kolmogorov–Smirnov (K–S) test.

**Human studies**. All human samples were collected after written informed consent by the patients and the study was approved by the review boards of three clinical entities: Humanitas Research Hospital (ID:1215), Verona AOUI University Hospital (ID:765cesc), and Ospedali Riuniti di Trieste (ID: 74/2015).

**Animal studies**. Animal experiments were limited to the teratoma formation assay to assess pluripotency of the generated iPSC lines and were performed in compliance with the national and EU guidelines for the care and use of laboratory animals. For the teratoma formation assay, six-week-old NOD:SCID mice (Charles River Laboratories) were used. The procedure was approved by the Italian Ministry of Health (137/2012-B).

**Reporting summary**. Further information on research design is available in the Nature Research Reporting Summary linked to this article.

## Data availability
All data that support the findings described in this study are available within the manuscript and the related supplementary information, and from the corresponding authors upon request. The source data underlying Figs. 1a–f, 2a–c, 3d, 4b, 7b and 7d, 8b–d, 9c-e and Supplementary Figs. 4D, 6, 7B–D, 12C–E and 13B–E are provided as a Source Data file.

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

## Acknowledgements

We thank Dr. G. Mostoslawsky (Boston University School of Medicine, USA) for providing the STEMCCA vectors for the reprogramming, Dr. Stephan Rohr (University of Bern) for the 22 mm coverslips with patterned growth strands used in the conduction velocity experiments and Dr. A. Brivanlou (Rockefeller University, NY, USA) for the RUES2 cell line. We are also grateful to Dr. D. Morone, Dr. Erreni and the Humanitas Imaging Unit for the assistance in the acquisition and analyses of the 3D-FISH and STED microscopy experiments; Drs. A. Anselmo and F.S. Colombo of the Flow Cytometry Core for the assistance with FACS; Roman Medvedev for the help in the conduction velocity experiments; Dr. P. Carullo for the teratoma formation assay; Drs. H. Nakahama and L. Rutigliano for the maintenance of the iPSC lines; and Dr. M. Rabino for generating the pLenti-LMNA-WT-GFP vector. This work was supported by the following grants: Italian Ministry of Health (GR-2011–02347743), Interomics Flagship Project of the CNR and "IO MERITO" Project of the Humanitas Research Center to E.D.P.; an EMBO Long-Term Fellowship (ALTF 448–2017) to A.B.; European Research Council Advanced Grant (CardioEpigen, # 294609), Italian Ministry of Health (PE-2013–02356818) and the CARIPLO Foundation (2015–0573) to G.C.; PNR-CNR Aging Program and Italian Ministry of Health (to PE-2011–02347329) to P.V.; Italian Ministry of Education, University and Research (2015583WMX) to G.C. and E.D.P.

## Author contributions

N.S. and S.C. performed most of the experiments, analysed data and wrote the manuscript. S.C. also contributed to generation of the isogenic control lines. M.M. contributed to functional experiments, analysed the data and contributed to writing the manuscript. A.B. designed the gene-editing strategy, generated isogenic control lines and contributed to the writing. M.P. performed the 3D-FISH experiments, analysed the data and contributed to writing. P.K. helped with the strategy for the chromatin immunoprecipitation experiments and the analyses. S.S. contributed to the protein distribution analysis of the 3D-STED experiments and assisted with the statistical analyses. A.F., C.L., G.S., M.D.F. and G.F. contributed to patients' samples collection and clinical evaluation. V.L. carried out the FACS analyses. P.V. contributed to the supervision of the 3D-FISH experiments and critically read the manuscript. C.E.M. supervised the generation of the gene-edited lines. N.S., S.C. and E.D.P. prepared the figures. G.C. contributed to interpretation of the results and wrote the manuscript. E.D.P. was responsible for forming the hypothesis, project development, data coordination, and writing, finalizing, and submitting the manuscript. All authors discussed and approved the manuscript.

## Additional information

**Competing interests:** The authors declare no competing interests.

