## [Peer Review File · Nature Communications]

Reviewers' comments:

Reviewer #1 (Remarks to the Author):

In this study Salvarani and colleagues presented some interesting evidence describing epigenetic alterations at the SCN5A gene in cardiomyopathic patients carrying the K219T mutation in Lamin A gene. The study was entirely conducted in iPS cells derived from patients and healthy controls as an in vitro model of human cardio-laminopathy. In the first set of data, using electrophysiological analyses in iPS-derived cardiomyocytes they showed reduced peak sodium currents and diminished conduction velocity. Then, they focalized their interest on the sodium channel Nav1.5, encoded by SCN5A gene. They found a consistent transcriptional reduction on SCN5A gene in patient-derived cardiomyocytes compared with controls. This reduction correlates with an increase of repressive histone markers and a decrease of active histone markers on SCN5A promoter. They found that the SCN5A locus is directly bound by Lamin A only in patient-derived cardiomyocytes. Same electrophysiological defects were observed in iPS-derived cardiomyocytes infected with mutated K219T form of Lamin A. Finally, the authors showed that remodelin-based treatments revert the electrophysiological alterations observed in patient-derived cardiomyocytes.

This work is potentially interesting, suggesting a key role for the mutated form of Lamin A (K219T) in the establishment of higher order chromatin structure that lead to an aberrant repression of the SCN5A locus. The understanding of the mechanism by which Lamin A-dependent higher order structures are folded and maintained in diseases is of great interest for the scientific community. I have some concerns about the extent to which the current molecular data support final conclusions. My general view is that authors, although nicely described the correlation between epigenetic features and transcriptional output, do not demonstrate that the higher order chromatin structure causes the aberrant repression. I recommend a major revision before publication in Nature Communication.

Major criticisms:

1. Figure 5: Authors measured some epigenetic marks, including the Lamin A interactions in control and patients-derived cardiomyocytes. The ChIP of Lamin A could be influenced by the presence of the mutated form of Lamin A. In fact, the mutated form of Lamin A could be more accessible to the antibody increasing the ChIP efficiency. In this view it is important to test the efficiency of endogenous Lamin A chromatin binding on some known targets and try to show, in non-saturated condition, that the antibody is working similarly with or without the mutated form of Lamin A.
2. Figure 5: The link of SCN5A locus and Lamin A in wt condition is not described. When repressed, does SCN5A locus localize in a LAD domain? It could be very informative to analyse the Lamin A binding during the wt cardiomyocytes differentiation. It is possible in fact that SCN5A locus is included in a LAD domain when repressed in wt undifferentiated cells and it loses contact with Lamin A during differentiation. Conversely in mutant cells this dynamics could be altered with mutated Lamin A retaining the binding with SCN5A locus. Similar dynamics were already showed in adipogenic differentiation (Lund, Collas Genome Research 2013) and can elucidate the mechanism at the basis of aberrant Lamin A binding.
3. The authors are proposing a model in which Lamin A recruits SCN5A locus in a PcG aggregate, close to the nuclear periphery. It is difficult to imagine how a single Lamin A point mutation changes the direct binding with the SCN5A locus. However, if this is the hypothesis, the wt vs mutated Lamin A interaction with SCN5A locus should be tested in vitro. Alternatively, if they hypothesize that mutated Lamin A interacts more with PcG proteins, they should test by co-IP the interaction between PcG and mutated/wt Lamin A.
4. Regarding the model described in the discussion section, authors stated: "In the mechanism we

propose here, mutant Lamin A/C proteins form a repressive complex with PRC2, which binds to the SCN5A gene promoter and catalyses the deposition of H3K27me3". Since, in their model, PRC2 play a key role in repressing SCN5A locus, the use of a specific PRC2 drug (such as DZNep) to revert the phenotype is strictly recommended.

5. Figure 9: This part of the work can be removed because is not in line with the rest of the paper. In the discussion section authors stated that" While the molecular mechanism by which Remodelin is acting in this context is still unknown". This is not correct since in 2014 Larrieu et al. in their Science publication described the target of Remodelin: the N-acetyltransferase 10 (NAT10). However, experiments showed by the authors are not complete, showing only the electrophysiological recovery and neglecting completely the molecular aspect.

Minor points:

- In the result section authors stated: "These results support the hypothesis that diminished sodium current density in LMNA-CMs is due to a reduction of SCN5A expression which, in turn, is mediated by enrichment on its promoter of Lamin A/C and the two repressive histone marks H3K9me3 and, particularly, H3K27me3". In addition about the LADs they comment: "Binding of Lamin A/C to chromatin at the nuclear periphery mostly occurs at variable LADs, which corresponds to LAD borders". These two statements are not precise: according to the last van Stenseel's review on Cell n 169 (Lamina-Associated Domains: Links with Chromosome Architecture, Heterochromatin, and Gene Repression") LADs are enriched in H3K9me3 and LADs boundaries are H3K27me3 enriched. LADs and LAD borders are not coinciding and present distinct epigenetic markers.
- Typos page 5 lane 99: Supplementary
- Which is the difference between R190W and K219T Lamin A mutations? Are the two mutations in the same protein domain? Please add a figure or explain in details.
- Figure 5: positive and negative controls of ChIP assay should be show for each antibody.
- I did not find any description of the Suz12 antibody used in the study

Reviewer #2 (Remarks to the Author):

This study focused on the role of K219T LMNA mutation in cardiac laminopathy using patient-specific iPSCs-CMs (LMNA-CMs). Various electrophysiological analyses showed LMNA-CMs have abnormal sodium currents with diminished conduction velocity. The authors suggest that altered epigenetic regulation in LMNA-CMs leads to down-regulation of SCN5A expression, resulting abnormal sodium currents. Although the findings presented in this manuscript are interesting, there are number of issues listed below.

Major issues

1. The authors need to specify the patient condition. Based on their previous paper (Ref 26), this family carried LMNA mutation as well as TNN mutation. So, authors need to provide the specific information including mutation, phenotype, age and gender of patients.
2. The author need to provide purity of CMs. The current protocol used in this study (Ref 23, and Ref 72), did not include purification step such as low glucose treatment. Although EP analysis is not likely to be significantly affected by purity of population, the general comparison analysis of protein (or mRNA) level will vary dependent on the purity of CMs.

3. There are critical statistic issues missing through all experiments. The authors claimed that they generated iPSCs lines from 3 patients and 2 healthy control. However, there is no detailed description of which line they used. For example, the authors compared EP parameters of control and patient iPSC-CMs. If 2 control and 3 patient lines were used for this, the authors need to provide appropriate indications of which line was used to generate each data point. If author used one line for each condition, the p-value could not be generated. For proper description, the author need to show individual (or average) data of each iPSC line.

4. Why LMNA A/C mutation selectively affects the transcriptional regulation of *NAV1.5*? As LMNA A/C contributes chromatin organization and transcriptional regulation of various genes, the additional experiments including RNA and ChIP-seq are needed to claim *SCN5A* as a major target of LMNA mutation.

5. Insufficient mechanism how LMNA mutation regulates *SCN5A* expression. Although the authors showed that *SCN5A* genomic region is highly located in nuclear periphery in LMNA-CMs, the precise mechanism how K219T mutation cause abnormal epigenetic modulation on *SCN5A* region is remained unclear.

6. The authors claimed that down-regulation of *SCN5A* expression could be a main mechanism of reduced sodium current density in LMNA-CMs. However, additional validations are needed to suggest this claim.

- Fig 4C: This figure showed that *SCN5A* is also localized in nuclei. To my knowledge, this protein mostly located in cytosolic fraction, especially in cell membrane (e.g., PMID: 28191886)

- Fig 4B: It is better to measure the band intensity. Based on B-actin level, control 1 likely showed similar or less *SCN5A* level of LMNA-CMs.

- Fig 6A and B: IF of *SCN5A* is not clear.

- The author also need to check whether overexpression of *SCN5A* rescues the phenotype of LMNA-CMs.

7. The author introduced mutant LMNA protein in hESCs-CMs to validate the effect of LMNA mutation on Na^+ currents and *SCN5A* level. For more accurate comparison, the authors need to compare GFP+ population of empty vector (only GFP) and K219T LMNA-GFP vector lines. Current experimental design could not exclude bias from virus infection. It is highly recommended to correct the mutation in LMNA lines using genome-editing method (e.g., CRISPR) to validate connection between LMNA mutation and phenotype of LMNA-CMs.

8. Did LMNA-CMs also have abnormal nuclear structure showed in previous paper (Fig 3E of Ref 26)? Could "Remodelin" treatment affect this phenotype? Moreover, the authors also need to measure the expression of *SCN5A* level in present of Remodelin.

9. The authors showed abnormal sarcomeric alignment in LMNA-CMs. How LMNA mutation contributes on this phenotype? Is there any possibility that this phenotype affects abnormal Na^+ currency? or conduct issues of patients? Did IF of patient tissues also show abnormal sarcomeric alignment (Fig 6A)?

Minor issues

1. LMNA R190W mutant line also showed the decrease of *SCN5A* level?

2. Is there any homozygous line carrying K219T LMNA mutation?

3. I believe the title is too strong. In addition, "epigenetic inhibition of sodium currents" may be inaccurate expression.

Reviewer #3 (Remarks to the Author):

Review

Salvarani, Crasto et al.

Lamin A/C mutations induce myocardial conduction defects through epigenetic inhibition of sodium currents in a human model of cardiac laminopathy.

The authors provide interesting data concerning the molecular and electrophysiological effects of a specific LMNA mutation in an iPSC-CM model. This data provides novel insight into the mechanism underlying the severe clinical phenotype observed in LMNA mutation carriers, which comprises not only of cardiomyopathy but is also characterised by a strong arrhythmogenic component. The techniques used, as well as the general conclusions concerning LMNA function that can be drawn from this data can be of interest to a broad audience of clinicians as well as basic scientists. However, in its current form the manuscript has some major drawback that need to be addressed as they prevent proper assessment of the data as essential controls are missing.

Major points:

- Two major effects of LMNA mutations described in the literature are not assessed in this paper:
 - o Macrostructural effects of the mutations on the CMs should be discussed and shown i.e. are there any effects on "blebbing"
 - o As apoptosis is one of the major effects described for LMNA mutations, it is important to measure the effects of the K219T mutation on apoptosis, this data is now conspicuously lacking in the manuscript
- Figure 1: data from stimulated action potentials needs to be presented as well in addition to the spontaneous APs. Including key characteristics describing the APs (e.g. not just upstroke velocity but also action potential duration etc)
- In figure 3 the authors show the data of conduction velocity experiments in 80um strands. This is very interesting data, however the upstroke velocity in these cells as described in suppl table 1 is too low to be mainly driven by NaV1.5. This is difficult to rhyme with the claim of the authors that the effect is largely through NaV1.5. It would be useful if a TTX experiment was performed (at least in the control line) demonstrating that the observed conduction velocity is indeed driven by NaV1.5
- The ChIP experiments suggest a higher level of interaction of LMNA K219T with the scn5a promoter region compared to WT (Figure 5) How do the authors consolidate this result with the fact that many patients have loss of function mutations in LMNA?
- In the experiments with the RUES2-CMs (figure 7) a proper control is lacking. At the least the effects of overexpression of the WT LMNA should be shown. Furthermore, the level of overexpression needs to be quantified.
- The data concerning the remodelin drug (figure 9) are very intriguing. However, key experiments and controls are missing. These results need further exploring to assess their relevance. At the moment this data is impossible to assess as the baseline effect of the drug on the cell type used is missing. What is the effect of remodelin on control CMs? Furthermore, these results pose more questions than they answer, mainly relating to how this drug affects the cardiomyocytes: does remodelin affect SCN5A expression levels? Are there structural changes in the cells studied as a result from the treatment? Does the drug affect NaV1.5 kinetics? Are there effects on the action potential? As this drug is largely unstudied in cardiomyocytes using it without proper controls (i.e. effects on

control cells) and without studying the effects in more detail does not add value to the paper.

Minor points

- Lee YK et al JAHA 2017 have published a study on the effects of two more mutations in iPSC-CMs this reference should be added and discussed in the introduction
- Proper gene and mutation nomenclature needs to be used throughout the manuscript: e.g. CX43 instead of Cnx43. What RefSeq does the K219T mutation refer to and what is the nucleotide change? There are more than 10 different protein coding transcripts of LMNA, this information is therefore crucial to be able to reproduce the data.
- The legend for the scale bars in Figure 2 A is missing
- In the legend of Figure 6B it is stated that the amount of red fluorescence is measured relative to the NaV1.5 protein level. What is meant by this? Were absolute levels measured by Western blot or is this a typo?
- Please provide the figures for the NaV1.5 kinetics in the wild type and mutant situation (this data is now only represented in the supplementary tables)
- In the methods section of the PAT-ChIP the amount of antibody used needs to be more specific than "the appropriate amount". Also mention the name of the software used for analysis.
- The title of the manuscript is too broad as this paper really only focusses on one mutation.

RESPONSE TO REFEREES

We thank the editor and the referee for their constructive comments and their positive feedback. The criticisms they raised and their recommendations helped us improve the quality of this manuscript. Below we reproduce in ***bold italics*** the referee's comments, followed by our response and how we have modified the manuscript when appropriate.

Replay referee #1

In this study Salvarani and colleagues presented some interesting evidence describing epigenetic alterations at the SCN5A gene in cardiomyopathic patients carrying the K219T mutation in Lamin A gene. The study was entirely conducted in iPS cells derived from patients and healthy controls as an in vitro model of human cardio-laminopathy. In the first set of data, using electrophysiological analyses in iPS-derived cardiomyocytes they showed reduced peak sodium currents and diminished conduction velocity. Then, they focalized their interest on the sodium channel Nav1.5, encoded by SCN5A gene. They found a consistent transcriptional reduction on SCN5A gene in patient-derived cardiomyocytes compared with controls. This reduction correlates with an increase of repressive histone markers and a decrease of active histone markers on SCN5A promoter. They found that the SCN5A locus is directly bound by Lamin A only in patient-derived cardiomyocytes. Same electrophysiological defects were observed in iPS-derived cardiomyocytes infected with mutated K219T form of Lamin A. Finally, the authors showed that remodelin-based treatments revert the electrophysiological alterations observed in patient-derived cardiomyocytes.

This work is potentially interesting, suggesting a key role for the mutated form of Lamin A (K219T) in the establishment of higher order chromatin structure that lead to an aberrant repression of the SCN5A locus. The understanding of the mechanism by which Lamin A-dependent higher order structures are folded and maintained in diseases is of great interest for the scientific community. I have some concerns about the extent to which the current molecular data support final conclusions. My general view is that authors, although nicely described the correlation between epigenetic features and transcriptional output, do not demonstrate that the higher order chromatin structure causes the aberrant repression. I recommend a major revision before publication in Nature Communication.

We thank the reviewer for the general positive impression on our manuscript and on the suggestions which we believe have improved the overall quality of the data.

Major criticisms:

1. Figure 5: Authors measured some epigenetic marks, including the Lamin A interactions in control and patients-derived cardiomyocytes. The ChIP of Lamin A could be influenced by the presence of the mutated form of Lamin A. In fact, the mutated form of Lamin A could be more accessible to the antibody increasing the ChIP efficiency. In this view it is important to test the efficiency of endogenous Lamin A chromatin binding on some known targets and try to show, in non-saturated condition, that the antibody is working similarly with or without the mutated form of Lamin A.

We have now included for the Lamin A/C-ChIP analysis three loci (*DEFA3*, *DEFA4* and *SCN10A*), which are known targets of Lamin A binding (Lund E. et al, 2013).

Results are shown in the new panel A of Figure 5, which demonstrates that Lamin A/C is highly enriched at the TTS of the *DEFA3*, *DEFA4* and *SCN10A* genes both in CNTR- and K219T-CM. Thus the possibility that ChIP itself could be influenced by the K219T mutation seems to be excluded, further supporting the involvement of K219T-mutated Lamin A/C in regulating *SCN5A* gene expression.

As requested by the reviewer, we have also included positive controls for the H3K27me3, H3K9me3, H3K4me3 and Suz12 antibodies: we used *NEUROD* for H3K27me3 and Suz12, *NANOG* for H3K9me3 and the cardiac-specific genes *TNNC1* and *ATCN2* for H3K4me3.

Data have been included in the new Figure 5 (panels B and C) and in the new supplementary figure 9 (panels A and B). We have modified the text accordingly (line 8-12 page 11).

2. Figure 5: The link of *SCN5A* locus and Lamin A in wt condition is not described. When repressed, does *SCN5A* locus localize in a LAD domain? It could be very informative to analyse the Lamin A binding during the wt cardiomyocytes differentiation. It is possible in fact that *SCN5A* locus is included in a LAD domain when repressed in wt undifferentiated cells and it loses contact with Lamin A during differentiation. Conversely in mutant cells this dynamics could be altered with mutated Lamin A retaining the binding with *SCN5A* locus. Similar dynamics were already showed in adipogenic differentiation (Lund, Collas Genome Research 2013) and can elucidate the mechanism at the basis of aberrant Lamin A binding.

We agree with the view proposed by the reviewer and, as suggested, we performed additional experiments to deepen the dynamics of *SCN5A* locus and wt Lamin A during differentiation of iPSC toward cardiomyocytes.

To answer the reviewer's question, we performed ChIP against Lamin A/C at different time points during cardiac induction (d0, d5, d8, d12) and results obtained are in line with the hypothesis raised by the reviewer, showing that the binding of Lamin A/C to *SCN5A* dynamically changes during differentiation. The results from these experiments are provided in the new supplementary figure 9, showing that Lamin A/C is enriched at *SCN5A* TSS in both CNTR and K219T-CMs until d5 of differentiation; an increased binding of Lamin A/C to *SCN5A* promoter persists at d8 of differentiation in mutant samples, while it is lost in wt cells, supporting an altered dynamic of LaminA/C binding to *SCN5A* gene during differentiation. Furthermore, this binding remains significantly higher in K219T-CMs compared to the control also at later stages and in terminally differentiated CMs, suggesting there is not just a delay in the Lamin A/C binding dynamic but more Lamin A/C is also bound at TTS of *SCN5A* in terminally differentiated CMs, consequently affecting the expression of the gene. Of note, the increased binding of Lamin A/C in adult myocardium suggested by PAT-ChIP experiments point toward a similar conclusion, which is that an increased binding of Lamin A/C to *SCN5A* locus is responsible for its decreased expression in K219T-mutated cells.

The text of the new version of the manuscript has been modified accordingly: line 2-8, page 13.

Regarding the reviewer's question on whether *SCN5A* localizes in a LAD in cardiomyocytes, unfortunately no "wet" data are available in the literature or by us; however, bioinformatic analyses conducted gathering data from cell types other than CMs (i.e., fibroblasts, HeLa cells, adipose stem cells) indicate that the *SCN5A* gene maps close to *SCN10A* and *SCN11A*, which are located in a LAD domain (Lund et al, 2013). As shown in the new Figure 5A, the analysis of the binding of Lamin A/C to *SCN10A* in CNTR and K219T-CMs revealed an enrichment in both conditions, regardless of the presence of the mutation, supporting a specific role of the mutant Lamin A/C on *SCN5A*, which is probably located in the variable portion of the LAD.

3. The authors are proposing a model in which Lamin A recruits *SCN5A* locus in a PcG aggregate, close to the nuclear periphery. It is difficult to imagine how a single Lamin A point

mutation changes the direct binding with the SCN5A locus. However, if this is the hypothesis, the wt vs mutated Lamin A interaction with SCN5A locus should be tested in vitro. Alternatively, if they hypothesize that mutated Lamin A interacts more with PcG proteins, they should test by co-IP the interaction between PcG and mutated/wt Lamin A.

The reviewer interpreted our hypothesis in exact terms. We agree with the reviewer that it is difficult to imagine that a single amino acid substitution can modify Lamin A/C binding to a genomic locus. As a matter of fact, our hypothesis was indeed that the mutation may alter the binding affinity to other chromatin-binding factors, including PcG. In order to demonstrate the latter and to deepen the understanding on the molecular mechanisms underlying our observations, we analysed LMNA-PRC2 protein-protein interaction using stimulated emission depletion (STED) super-resolution microscopy. The colocalization analysis showed higher levels of the interaction between Lamin A/C and SUZ12 (a PRC2 subunit) in K219T-LMNA-CMs compared to the control, measured as number of colocalized voxels. Results from these experiments have been included in the new Figure 5, as panel D and discussed in the text at page 12, line 14-17.

4. Regarding the model described in the discussion section, authors stated: “In the mechanism we propose here, mutant Lamin A/C proteins form a repressive complex with PRC2, which binds to the SCN5A gene promoter and catalyses the deposition of H3K27me3”. Since, in their model, PRC2 play a key role in repressing SCN5A locus, the use of a specific PRC2 drug (such as DZNep) to revert the phenotype is strictly recommended.

We agree with the reviewer that a selective inhibition of PRC2 complex would further support the mechanism we are describing here. However, when we tried to perform the experiment suggested by the reviewer using DZNep, the treatment resulted to be highly toxic for differentiated CMs, with a percentage of mortality was higher than 50% at all concentrations of drug we tested (0.5µM; 1µM; 2 µM; 5 µM).

We do hope that the reviewer will be convinced by the large amount of data included in the new version of the manuscript, among which those related to LaminA/C-PRC2 colocalization as well as those gathered from the isogenic “corrected” lines (see below).

5. Figure 9: This part of the work can be removed because is not in line with the rest of the paper. In the discussion section authors stated that” While the molecular mechanism by which Remodelin is acting in this context is still unknown”. This is not correct since in 2014 Larrieu et al. in their Science publication described the target of Remodelin: the N-acetyltransferase 10 (NAT10). However, experiments showed by the authors are not complete, showing only the electrophysiological recovery and neglecting completely the molecular aspect.

While we understand the request of the reviewers and agree that in the cited report it is described a mechanism for the action of the chemical on the nuclear morphology, we are uncertain that the drug could regulate also the expression of our target gene with the same mechanism.

[Redacted]

[Redacted]

Given the potential therapeutic relevance of this molecule for treatment of laminopathies, we believe that the subject deserves further investigation.

In light of all these considerations, we concur with reviewer 1 that this part of the manuscript can be removed from the text, becoming the starting point of another study.

Minor points:

- In the result section authors stated: "These results support the hypothesis that diminished sodium current density in LMNA-CMs is due to a reduction of SCN5A expression which, in turn, is mediated by enrichment on its promoter of Lamin A/C and the two repressive histone marks H3K9me3 and, particularly, H3K27me3". In addition about the LADs they comment: "Binding of Lamin A/C to chromatin at the nuclear periphery mostly occurs at variable LADs, which corresponds to LAD borders". These two statements are not precise: according to the last van Stenseel's review on Cell n 169 (Lamina-Associated Domains: Links with Chromosome Architecture, Heterochromatin, and Gene Repression") LADs are enriched in H3K9me3 and LADs boundaries are H3K27me3 enriched. LADs and LAD borders are not coinciding and present distinct epigenetic markers.

We agree with the reviewer and rephrased the sentence in the text (page 12, line 20-22) as follow: “LADs are mostly cell-specific and may be distinguished into constitutive (cLADs) and variable (vLADs), the latter corresponding to LAD borders ...”

- Typos page 5 lane 99: Supplementary

We have corrected the typos in the new revised version

- Which is the difference between R190W and K219T Lamin A mutations? Are the two mutations in the same protein domain? Please add a figure or explain in details.

R190W and K219T mutations are located in exons 3 and 4 of the *LMNA* gene respectively; at the protein level, both mutations are located in the rod domain, with the former falling in the Coil 1B region, and the latter being the first amino acid of the linker region (between Coil 1B and Coil 2). We added a panel highlighting the position of the mutations in the Supplementary Figure 2 and 6. We also specified this in the text at page 9, line 20-22.

- Figure 5: positive and negative controls of ChIP assay should be show for each antibody.

As requested, we included controls for the H3K27me3, H3K9me3 and H3K4me3 antibodies in the new panel B of Figure 5 and in the new supplementary figure 9 (panels A and B).

- I did not find any description of the Suz12 antibody used in the study

The antibody used in the study for Suz12 is the following: Cell Signaling, #3737. We included the details in the Supplementary Methods.

Reviewer #2

This study focused on the role of K219T LMNA mutation in cardiac laminopathy using patient-specific iPSCs-CMs (LMNA-CMs). Various electrophysiological analyses showed LMNA-CMs have abnormal sodium currents with diminished conduction velocity. The authors suggest that altered epigenetic regulation in LMNA-CMs leads to down-regulation of SCN5A expression, resulting abnormal sodium currents. Although the findings presented in this manuscript are interesting, there are number of issues listed below.

We thank reviewer #2 for the criticisms; by responding to them, we have certainly improved the quality of our work.

Major issues

1. The authors need to specify the patient condition. Based on their previous paper (Ref 26), this family carried LMNA mutation as well as TNN mutation. So, authors need to provide the specific information including mutation, phenotype, age and gender of patients.

We have generated iPSCs from 3 familial patients affected by dilated cardiomyopathy and carrying K219T LMNA mutation (LMNA #1, LMNA #2 and LMNA #3), and from 1 isolated case diagnosed with dilated cardiomyopathy and carrying the R190W mutation in the LMNA gene. LMNA #2 and #3 are also carriers of an additional modifier variant in the TTN gene (p.L4855F). However, at least with regard to the comprehensive electrophysiological analysis performed in this study, we did not observe any difference among CMs obtained from patients carrying the additional TTN variant and those with LMNA mutations only. More relevantly, any potential effect of TTN gene variant on the phenotype is excluded by virtue of the new results generated from CMs from K219T-corrected isogenic control lines, which demonstrate the causal relationship between the LMNA K219T mutation and the sodium current phenotype. Patients' information are listed in Supplementary Table 1 and included in the supplementary material in the Extended Methods section (page 2, line 3-12)

2. The authors need to provide purity of CMs. The current protocol used in this study (Ref 23, and Ref 72), did not include purification step such as low glucose treatment. Although EP analysis is not likely to be significantly affected by purity of population, the general comparison analysis of protein (or mRNA) level will vary dependent on the purity of CMs.

We agree with the reviewer's criticism and are aware of issues related to the heterogeneity of CMs derived from iPSC differentiation. Experiments using cell lines corrected by genome editing should clarify doubts related to the cell population purity. Following reviewer's recommendations, we nonetheless added a FACS analysis that assesses the purity of CMs using the cardiac-specific marker alpha-sarcomeric actinin. For all clones, the efficiency in generating cardiac myocytes was reproducibly higher than 75% (in a range from 75 to 95% cells positive for the α -sarcomeric actinin cardiac specific marker). Results of these experiments have been included in the Supplementary Figure 3 (panel A) and the methodology added to the Material and Methods section (page 3 line 14-24).

3. There are critical statistic issues missing through all experiments. The authors claimed that they generated iPSCs lines from 3 patients and 2 healthy control. However, there is no detailed description of which line they used. For example, the authors compared EP parameters of control and patient iPSC-CMs. If 2 control and 3 patient lines were used for this, the authors need to provide appropriate indications of which line was used to generate each data point. If author used one line for each condition, the p-value could not be generated. For proper

description, the authors need to show individual (or average) data of each iPSC line.

We thank the reviewer for raising this criticism. In general, experiments were performed on two independent clones for each individual and data are presented as average of all the lines for the two main conditions (CNTR and K219T), as stated in the Extended Methods section of the Supplementary Material (pag 2 – line 21-22). We indeed missed to highlight it in the main text.

In the revised version, we have included this specific information also in the main text (in the Material and Methods section, pag. 21, line 7-8).

Values obtained in the experiments, both functional and molecular, are average for all independent lines (either controls or mutated – two lines for each patient).

In order to satisfy the reviewer's criticism, we have highlighted in a separate graph in the supplementary material the single values of the maximal diastolic potential (MDP) measured in the individual lines (see new Supplementary Figure 4). We chose to show data from individual CM lines for this specific parameter because it represents the electrophysiological recording from which all the other AP parameters were measured.

4. Why LMNA A/C mutation selectively affects the transcriptional regulation of Nav1.5? As LMNA A/C contributes chromatin organization and transcriptional regulation of various genes, the additional experiments including RNA and ChIP-seq are needed to claim SCN5A as a major target of LMNA mutation.

We agree with the reviewer that it would be very interesting to investigate the transcriptional effect of the Lamin A/C mutation through the ChIP approach on the whole genome; however, these investigations fall outside the scope of this manuscript. We do not claim that *LMNA* mutation is acting selectively on $Na_v1.5$; on the contrary, a modulating effect also on other genes is to be expected. Indeed, preliminary RNA-sequencing data we obtained on CNTR- and K219T-CMs indicated a variety of genes differentially modulated in the two cellular models (not-shown). Of particular relevance, genes of the cardiac conduction system (including *SCN5A*) and of contractility are included among those differentially regulated in the two conditions. The reason why we focused on *SCN5A* gene is because EP experiments indicated that the sodium current, and therefore $Na_v1.5$, was grossly altered compared to other currents. We therefore hypothesized that alteration of *SCN5A* gene expression could be a leading mechanism of the defects in cardiac conduction and excitability induced by K219T *LMNA* mutation.

5. Insufficient mechanism how LMNA mutation regulates SCN5A expression. Although the authors showed that SCN5A genomic region is highly located in nuclear periphery in LMNA-CMs, the precise mechanism how K219T mutation cause abnormal epigenetic modulation on SCN5A region is remained unclear.

In order to deepen the molecular mechanisms behind the epigenetic modulation by Lamin A/C on the *SCN5A* region and reply to this criticism, we extended our study and tested whether this could be mediated by its interaction with PRC2 complex. We thus analysed the Lamin A/C-PRC2 protein-protein interaction using stimulated emission depletion (STED) super-resolution microscopy. As already included in the response to reviewer #1, the colocalization analysis showed that the interaction between Lamin A/C and SUZ12, a PRC2 subunit, was higher in K219T-CMs compared to the control, expressed as number of co-localized voxels. Results from these experiments have been included in the manuscript (page 12 line 14-17) and are shown in the new Figure 5, panel D.

6. The authors claimed that down-regulation of SCN5A expression could be a main mechanism of reduced sodium current density in LMNA-CMs. However, additional validations are needed to suggest this claim.

- Fig 4C: This figure showed that SCN5A is also localized in nuclei. To my knowledge, this protein mostly located in cytosolic fraction, especially in cell membrane (e.g., PMID: 28191886)

We agree with the reviewer that Na_v1.5 protein should be located at the plasma membrane; however, a nuclear localization of this protein has been recently reported (Onwuli DO et al, Channels vol.11 – 2017). More in detail, a nuclear localization signal (NLS) was found inside the linker between two of the four homologous domains which Na_v1.5 is made up of (L_{DI-DII}), supporting localization of sodium channel proteins not exclusively in the cytosol/plasma membrane compartments and justifying the nuclear signal detected in immunofluorescence experiments reported here.

- Fig 4B: It is better to measure the band intensity. Based on B-actin level, control 1 likely showed similar or less SCN5A level of LMNA-CMs.

As suggested by the reviewer, we quantified protein levels by measuring band intensity from the Western blot experiments. Results have been added to the new Figure 4 as panel B.

- Fig 6A and B: IF of SCN5A is not clear.

We hope we correctly interpreted what was not clear to the reviewer. In order to improve the IF of Na_v1.5, we have replaced the IF for sarcomeric actinin (in green in the original version of the figure), which suffered from some green autofluorescence coming from the paraffin in the sample, with phase-contrast images of the heart section.

The new panels have been included in the new Figure 6.

The authors also need to check whether overexpression of SCN5A rescues the phenotype of LMNA-CMs.

In order to answer the reviewer's comment, we have generated a vector expressing a GFP-tagged SCN5A (similarly to what has been described for overexpressing the mutant LMNA) and performed electrophysiological studies on mutant cells (K219T-LMNA-CMs) exposed to lentiviral particles expressing the SCN5A-GFP fusion protein.

Results from this set of experiments showed a complete rescue of the phenotype, in terms of currents, with even higher intensities in mutant CMs transduced with SCN5A compared to the controls. We also determined that overexpression of SCN5A gene exerted a positive effect also on the dV/dt_{max} and on the other related AP parameters, which are all significantly increased in comparison to K219T-LMNA-CMs, as shown in the new Supplementary Table 5. Data are from both spontaneously active and evoked APs.

Similar results were obtained in CMs carrying the R190W mutation.

Results from these SCN5A-overexpression experiments have been included in the new Supplementary Figure 11, and detailed AP parameters are provided in the supplementary Table 5. Text has been modified accordingly (page 14 line 19-25; page 15 line 1-3).

7. The author introduced mutant LMNA protein in hESCs-CMs to validate the effect of LMNA mutation on Na⁺ currents and SCN5A level. For more accurate comparison, the authors need to compare GFP+ population of empty vector (only GFP) and K219T LMNA-GFP vector lines. Current experimental design could not exclude bias from virus infection. It is highly recommended to correct the mutation in LMNA lines using genome-editing method (e.g., CRISPR) to validate connection between LMNA mutation and phenotype of LMNA-CMs.

We have followed reviewer's suggestions: in the revised version, not only have we provided additional controls for the overexpression experiments in hESC-CMs, but, importantly and

tiresomely, we generated isogenic lines with CRISP/Cas9 technology. For overexpression experiments, we performed all the additional controls suggested by the reviewer and analysed the sodium current density in RUES2-derived CMs transduced with either GFP empty vector or WT-LMNA-GFP. In both cases, registered values were similar to those obtained in the GFP^{neg} (RUES2 CMs, not carrying the mutation) and CNTR-CMs, suggesting that the lentiviral transduction did not lead to any bias.

Results from these experiments have been included in the supplementary figure 10, and registered parameters are listed in the Supplementary Table 4. We modified the text of the manuscript accordingly (page 14 line 15-18).

To further corroborate our data and unequivocally demonstrate the effect of the K219T *LMNA* mutation, we also generated isogenic lines from the *LMNA* #1_2 mutant line and analysed CMs differentiated from two independent clones. For time constraints, we chose to perform experiments aimed at determining whether genotype reversion normalized sodium currents, the expression of *SCN5A* gene and its binding to Lamin A/C.

Results from these experiments have been included in the manuscript as a new figure 8, showing that sodium currents in “corrected lines” are 1) comparable to control cells and 2) result to be significantly higher than currents recorded in the parental K219T-CMs, supporting the specific action of the mutation on sodium currents. Consistently, we found that *SCN5A* expression was restored to control levels in “corrected” CMs while binding of Lamin A/C to its promoter was significantly decreased compared to K219T mutant cells. Results of the experiments on the isogenic control lines are shown in the new Figure 8 and added in the text at page 15 (line 4-16).

Experimental details on generation and characterization of the isogenic lines are provided in the supplementary material (in the Extended Methods section, pages 7-10) and in the Supplementary Figure 12.

We hope the reviewer appreciates the enormous effort needed to generate the isogenic lines.

8. Did *LMNA*-CMs also have abnormal nuclear structure showed in previous paper (Fig 3E of Ref 26).

While fibroblasts from patients’ skin biopsies as well as cardiomyocytes in heart tissue sections did show nuclear abnormalities typical of laminopathic cells, we couldn’t detect a significant number of cells with nuclear structure abnormality in K219T-CMs *in vitro*.

We analysed 65 and 58 nuclei of control and K219T CMs, respectively, generated from various lines and found no significant differences in the percentage of cells displaying nuclear abnormalities in both sample groups. In detail, only 6 CNTR-CMs (out of 65) and 9 K219T-CMs (out of 58) presented with nuclear abnormalities.

An explanation of this phenomenon could only be a subject of speculation at this point. It is for instance possible that nuclear structure could become evident only by growing CMs in more physiological conditions, such as a 3D, organoid-like ones. Alternatively, it is possible that long term culture, such as that used for generating cardiomyocytes, which typically lasts more than 35 days, could influence nuclear architecture.

Data have been included in the supplementary figure 3 (panel E) and discussed in the text (page: 6 line: 21-25).

Could “Remodelin” treatment affect this phenotype? Moreover, the authors also need to measure the expression of *SCN5A* level in the presence of Remodelin.

Since no nuclear abnormalities were detected, we were unable to test the effect of Remodelin on this phenotypic trait;

[Redacted]

9. The authors showed abnormal sarcomeric alignment in LMNA-CMs. How LMNA mutation contributes on this phenotype? Is there any possibility that this phenotype affects abnormal Na currency? or conduct issues of patients? Did IF of patient tissues also show abnormal sarcomeric alignment (Fig 6A)?

In this study we show that LMNA-CMs carrying the K219T mutation display sarcomeric disorganization and increased size (Supplementary Figure 3). How the LMNA mutation specifically contributes to this phenotype was not a major point of our study. Those data were included to demonstrate that the generated models recapitulate morphological characteristic of the disease.

However, we can assume that the abnormal sarcomeric alignment observed in our models is driven by other pathways, such as MAPK/ERK pathway, which has already been shown involved in regulating sarcomere structure (Muchir A et al, 2007, Chatzifrangkeskou M et al, 2018). On this regard, we have recently contributed to the work of Chatzifrangkeskou M et al, showing that ERK1/2-mediated phosphorylation of the actin depolarizing factor cofilin-1 leads to disassembly of actin filaments both *in vivo* and *in vitro* in several cellular models of LMNA-cardiomyopathy, including iPSC-CMs carrying the R190W mutation.

We don't have any data supporting the notion that sarcomeric disorganization affects sodium current or leads to conduction defects in patients; however, experiments from isogenic lines demonstrated a direct causal effect of the mutation on the regulation of *SCN5A* expression, the gene encoding the sodium channel, supporting the hypothesis that defects of conduction are instead a direct consequence of the reduction of sodium currents.

Regarding the question of whether IF revealed abnormal sarcomeric alignment on patient tissues, we couldn't detect any dramatic effect on sarcomere structure in the samples analysed in this study; however, the performed analyses were not intended to detect ultrastructural defects, and we may have missed this phenotypic feature.

Minor issues

1. LMNA R190W mutant line also showed the decrease of SCN5A level?

Yes. We have added a gene expression panel in the supplementary figure 6, showing that also R190W *LMNA* mutation induces a decrease in *SCN5A* gene expression (panel E). We modified the text accordingly (page: 10 line 25: page 11 line 1).

2. Is there any homozygous line carrying K219T LMNA mutation?

No, there are no reported cases of homozygous K219T-LMNA mutations or any homozygous lines carrying the K219T mutation. Laminopathies are autosomal dominant diseases. Indeed, most Lamin A/C mutations are heterozygous; homozygous mutation of *LMNA* gene have been described in rare cases associated with the most severe HGPS and MAD forms.

3. I believe the title is too strong. In addition, “epigenetic inhibition of sodium currents” may be inaccurate expression.

Following the suggestion of the reviewer, we changed the title as follow: “K219T-Lamin A/C mutation induces myocardial conduction defects through epigenetic inhibition of *SCN5A* in a human model of cardiac laminopathy”

Reviewer #3

The authors provide interesting data concerning the molecular and electrophysiological effects of a specific LMNA mutation in an iPSC-CM model. This data provides novel insight into the mechanism underlying the severe clinical phenotype observed in LMNA mutation carriers, which comprises not only of cardiomyopathy but is also characterised by a strong arrhythmogenic component. The techniques used, as well as the general conclusions concerning LMNA function that can be drawn from this data can be of interest to a broad audience of clinicians as well as basic scientists. However, in its current form the manuscript has some major drawback that need to be addressed as they prevent proper assessment of the data as essential controls are missing.

Major points:

- Two major effects of LMNA mutations described in the literature are not assessed in this paper:

o Macrostructural effects of the mutations on the CMs should be discussed and shown i.e. are there any effects on “blebbing”

We thank the reviewer for the observation that gave us the opportunity to discuss further this issue.

By immunofluorescence, fibroblasts clearly show nuclear abnormalities (i.e. blebbing and invagination) typical of laminopathic cells (a representative figure is included here for the reviewers, as Figure 2); on the contrary, from a retrospective analysis of immunofluorescence studies on iPSC-CMs, such macrostructural defect did not emerge (new supplementary fig. 3E). As already mentioned in the reply to reviewer #2 (please read above), we do not have an experimental explanation for this observation, but only speculative ones.

Figure 2 - Nuclear morphology of primary fibroblasts from K219T-LMNA patients. Representative images of Lamin A/C immunostaining in fibroblasts from control subjects (CNTR) and patients carrying the K219T mutation (LMNA #1 and LMNA #2), showing abnormal nuclear morphology in mutant cells, with nuclear membrane invagination and lobulation.

o As apoptosis is one of the major effects described for LMNA mutations, it is important to measure the effects of the K219T mutation on apoptosis, this data is now conspicuously lacking in the manuscript

We are aware that, in particular under stress conditions, *LMNA* mutations have a major effect on apoptosis. However, since we didn't notice any macroscopic effect on cell death in our cellular models, we didn't investigate it further. In fact, results from FACS experiments on live/dead-stained CMs (see Figure 3 of this reply to reviewers) show a percentage of death of approximately 10% in each line from CNTR and K219T LMNA-CMs in basal conditions, suggesting that no major effect on apoptosis is induced by the K219T mutation in CMs, at least in basal conditions.

Figure 3 - Flow cytometry analysis of cell viability of iPSC-derived CMs. Left: representative flow cytometry plots of human iPSC-CMs. Debris were excluded and cells were selected based on size (upper left). After doublets (upper right), cells were analysed using the Live/Dead Fixable Aqua Stain (ThermoScientific), that specifically stains dead cells. Scatter plots and gates of a representative experiment are shown in the bottom left panels. Right: percentage of live cells. Histograms depict all analysed clones in duplicate.

- Figure 1: data from stimulated action potentials needs to be presented as well in addition to the spontaneous APs. Including key characteristics describing the APs (e.g. not just upstroke velocity but also action potential duration etc)

As requested by the reviewer, data from stimulated action potential are now introduced as graphs in the new figure 1 of the manuscript, as well as APDs parameters.

These data are also available in the Supplementary Tables 2 and 3.

- In figure 3 the authors show the data of conduction velocity experiments in 80um strands. This is very interesting data, however the upstroke velocity in these cells as described in suppl table 1 is too low to be mainly driven by NaV1.5. This is difficult to rhyme with the claim of the authors that the effect is largely through NaV1.5. It would be useful if a TTX experiment was performed (at least in the control line) demonstrating that the observed conduction velocity is indeed driven by NaV1.5

Following the reviewer’s suggestion, we performed additional experiments accordingly. Results are available in the supplementary section as Supplementary Figure 7 and in the text (page 10, line 12-14). In brief, treatment with TTX completely abolished conduction multicellular-strand of control CMs, indicating that the conduction velocity measured in our samples is mainly driven by Na_v1.5.

- The ChIP experiments suggest a higher level of interaction of LMNA K219T with the scn5a promoter region compared to WT (Figure 5) How do the authors consolidate this result with the fact that many patients have loss of function mutations in LMNA?

LMNA mutations may act either by aploinsufficiency or by a dominant-negative mechanism. In our study, we focused principally on one specific mutation (K219T); results support a dominant-negative mechanism of the mutation.

- In the experiments with the RUES2-CMs (figure 7) a proper control is lacking. At the least the

effects of overexpression of the WT LMNA should be shown. Furthermore, the level of overexpression needs to be quantified.

According to the reviewer's suggestion, we performed additional overexpression experiments in RUES2-CMs, using an empty-GFP vector and one expressing the WT *LMNA* gene. Results of these experiments, provided as supplemental material in the Supplementary Figure 10, show that sodium current densities in cells in which GFP empty vector or WT *LMNA* are overexpressed are indistinguishable from the GFP^{neg} population (control CMs that do not express any vector), indicating that the vector itself does not interfere with the sodium current densities. Regarding the quantification of overexpression, FACS analysis data from sorting experiments (Figure 7E) indicates that approximately 30% of the cells were positive for GFP expression.

Of note, the experiments were carried out either on single cells (GFP positive) or on GFP-positive sorted cell populations.

- The data concerning the remodelin drug (figure 9) are very intriguing. However, key experiments and controls are missing. These results need further exploring to assess their relevance. At the moment this data is impossible to assess as the baseline effect of the drug on the cell type used is missing. What is the effect of remodelin on control CMs? Furthermore, these results pose more questions than they answer, mainly relating to how this drug affects the cardiomyocytes: does remodelin affect SCN5A expression levels? Are there structural changes in the cells studied as a result from the treatment? Does the drug affect Nav1.5 kinetics? Are there effects on the action potential? As this drug is largely unstudied in cardiomyocytes using it without proper controls (i.e. effects on control cells) and without studying the effects in more detail does not add value to the paper.

Ex-post, we agree with the reviewer's criticisms: initially, we thought that including pharmacological data would confer more interest to the manuscript, considering the potential therapeutic implication of Remodelin in cardiac laminopathies.

[Redacted]

[Redacted]

[Redacted]

Based on these new evidence and following also reviewer #1's suggestion, we believe this part of the manuscript can be removed from the text. It will provide to us the basis for further studies on this topic.

We will include these data in the manuscript at the reviewers' request.

Minor points

- Lee YK et al JAHA 2017 have published a study on the effects of two more mutations in iPSC-CMs this reference should be added and discussed in the introduction

We include the reference in the new version of the manuscript

- Proper gene and mutation nomenclature needs to be used throughout the manuscript: e.g. CX43 instead of Cnx43. What refseq does the K219T mutation refer to and what is the nucleotide change? There are more than 10 different protein coding transcripts of LMNA, this information is therefore crucial to be able to reproduce the data.

We went through the manuscript and fixed the proper gene and mutation nomenclature. The Refseq for *LMNA* gene and the specific nucleotide change are provided in the figure legend of the Supplementary Figure 2 (page 20, line 4-12 of the supplementary information)

- The legend for the scale bars in Figure 2 A is missing

We have now added the legend for the scale bars in Figure 2 A, as requested.

- In the legend of Figure 6B it is stated that the amount of red fluorescence is measured relative to the Nav1.5 protein level. What is meant by this? Were absolute levels measured by Western blot or is this a typo?

Figure 6B referred to the quantification of the level of red fluorescent signal, which corresponds to Nav1.5 protein in the IF staining shown in the panel A. For quantification, ImageJ Fiji software was used.

- Please provide the figures for the Nav1.5 kinetics in the wild type and mutant situation (this data is now only represented in the supplementary tables)

Voltage dependencies for Nav1.5 are now graphically introduced for all tested conditions.

- In the methods section of the PAT-ChIP the amount of antibody used needs to be more specific than “the appropriate amount”. Also mention the name of the software used for analysis.

In the PAT-ChIP experiment, 3 µg of anti-Lamin A/C (ChIP grade - sc-7292) were used.

qPCR analyses were performed with Viia7 software and relative enrichment was calculated as ChIP/input ratio.

- The title of the manuscript is too broad as this paper really only focusses on one mutation.

According to reviewer’s suggestion, the title has been modified as follow: “K219T-Lamin A/C mutation induces myocardial conduction defects through epigenetic inhibition of *SCN5A* in a human model of cardiac laminopathy”

Cited references

1. Lund E, Oldenburg AR, Delbarre E, Freberg CT, Duband-Goulet I, Eskeland R, et al. Lamin A/C-promoter interactions specify chromatin state-dependent transcription outcomes. *Genome Res.* 2013;23(10):1580-9.
2. Balmus G, Larrieu D, Barros AC, Collins C, Abrudan M, Demir M, et al. Targeting of NAT10 enhances healthspan in a mouse model of human accelerated aging syndrome. *Nat Commun.* 2018;9(1):1700.
3. Onwuli DO, Yanez-Bisbe L, Pinsach-Abuin ML, Tarradas A, Brugada R, Greenman J, et al. Do sodium channel proteolytic fragments regulate sodium channel expression? *Channels (Austin).* 2017;11(5):476-81.
4. Muchir A, Pavlidis P, Decostre V, Herron AJ, Arimura T, Bonne G, et al. Activation of MAPK pathways links LMNA mutations to cardiomyopathy in Emery-Dreifuss muscular dystrophy. *J Clin Invest.* 2007;117(5):1282-93.
5. Chatzifrangkeskou M, Yadin D, Marais T, Chardonnet S, Cohen-Tannoudji M, Mougenot N, et al. Cofilin-1 phosphorylation catalyzed by ERK1/2 alters cardiac actin dynamics in dilated cardiomyopathy caused by lamin A/C gene mutation. *Hum Mol Genet.* 2018.

Reviewers' comments:

Reviewer #1 (Remarks to the Author):

In the revised version of this manuscript authors performed a huge amount of additional experiments, improving the quality of the work.

I am satisfied of how they address my concerns, however I have still some comments on the representation and interpretation of new data.

Figure 5A and Supplementary Figure 9:

I noticed that the ChIP experiments were represented differently, In Figure 5A as % of the input, in supplementary Figure 9 as fold change over the IgG. ChIP experiments should be comparable along the manuscript and preferably showed as % of input. In line with this observation, the binding measured at d0 in wt cells on SCN5A locus (Supplementary figure 9) should be similar to the one measured in Figure 5A. Why they are so different? Does data representation influence the result? Are they using different primer pairs (and if so, why?). Are iPS conditions different? If so I would add the condition presented in Figure 5A in the Supplementary figure 9 graph.

If the authors can confirm this different dynamic of Lamin A binding at the SCN5A locus this panel should be placed in the Figure 5, because it represents an important piece of information about the molecular mechanism.

Figure 5D and E:

- In the revised version of the manuscript, authors, for the first time hypothesized a direct, physical interaction between mutated Lamin A and PRC2. I believe, also looking at new experiments that this is the right direction. However, in the text this is not clearly stated and FISH experiments showing a peripheral localization in K219T cells (Figure 5E) were not commented in the context of Lamin A-PRC2 physical interaction. If their hypothesis consider an aberrant and constitutive interaction between K219T Lamin with PRC2, it is possible to suppose that PRC2, already bound to the SCN5A locus is retained at the nuclear periphery, tethering the locus in a more repressive environment? Do I correctly interpret what they want to claim? In this case STED analysis should be repeated dividing nuclei in distinct zones (central and peripheral). If K219T-Lamin is retaining PRC2 in the nuclear periphery, analysis performed in the periphery zone should reveal a difference between wt and K219T-mutant more consistent than one performed in the central zone.

In addition, quantitative analysis of SUZ12 immunofluorescence dividing nuclei in distinct zones (central and peripheral) could show a relocalization of PRC2 in the K219T mutant.

- Representative images from wt and mutant are recommended.

- Statistic analysis of graph in Figure 5D is not convincing. How many measurements they performed? How many independent clones? Which cells were used? Which statistic analysis was used?

- Authors wrote in the figure legend: "On the right, a representative image of a nucleus stained for the two proteins" but the image is on the left.

Overall, I suggest to re-analyse some data and to add few experiments to better support their working hypothesis. In addition, the proposed molecular mechanism should be described clearly along the text (abstract, result and discussion).

--

Reviewer #2 (Remarks to the Author):

Major comments

I appreciate the authors' efforts to address my previous comments, especially generation of additional iPSC lines (Fig. 7). The current revised manuscript showed increase quality and quantity of data that support below two main points of this manuscript. I have no additional questions for these main points:

- 1) Lamin A/C mutation (K219T) induces abnormal sodium current of mutant iPSC-CMs.
- 2) Decreased SCN5A is required for disease phenotype in mutant iPSC-CMs.

However, it is still difficult for me to be convinced that lamin A/C mutation decreases the expression of SCN5A gene in mutant iPSC-CMs through epigenetic modulation, which could be essential molecular mechanism throughout the entire manuscript. The main question is how lamin A/C mutation (K219T) affects the transcriptional regulation of SCN5A in mutant iPSC-CMs.

In this revised manuscript, the author showed that lamin A/C and Suz12 (PRC2 complex protein), is highly enriched in the promoter region of SCN5A gene (Fig. 5C). The authors also claimed that mutant lamin A/C protein has higher binding affinity with Suz12 (Fig. 5D). In this point, I am confused what the suggested mechanism for the abnormal expression of SCN5A gene in mutant iPSC-CMs is. Do these data mean that lamin A/C mutation induces global alteration of PRC2 composition and that abnormal PRC2 occupancy recruits lamin A/C into the promoter region of SCN5A?? Or Lamin A/C mutation results in new LADs including SCN5A region and additionally recruits PRC2 complex into the promoter region of SCN5A??

First, if authors want to claim that mutant lamin A/C has more binding affinity with Suz12, then additional experiment such as immuno-precipitation (IP) is needed to check the difference in the binding capacity between wild-type lamin A/C and mutant lamin A/C.

Second, if authors want to claim that there is alteration of LADs in mutant iPSC-CMs compared to control iPSC-CMs, then global occupancy of lamin A/C on the genome should be analyzed. Because the mean size of LAD is 1-2MB (2016 Gesson et al. Genomic Research), there is limitation to identify the alteration of LADs using ChIP-qPCR.

Regardless of global distribution of lamin A/C, what is the possible mechanism that mutation (K219T) on the lamin A/C leads to abnormal occupancy of lamin A/C on the promoter region of SCN5A?

Above all, these data suggest that the alteration of either LADs or binding affinity to PRC2 complex will result in global gene expression changes. Again, I highly speculate that not only SCN5A gene but also the combination of various genes should contribute to the disease phenotype of mutant iPSC-CMs. I think the authors need to clarify their hypothesis and reinforce it through additional results.

Minor comments

1. To my knowledge, in hESCs (or iPSCs), the lamin B1 is highly expressed, but lamin A/C protein level is very low. How could the authors perform ChIP with native lamin A/C in D0 of CM differentiation (Supplementary Fig. 9C)? Did authors test the protein level of lamin A/C during CM differentiation and compare it in control and mutant lines?

2. Previous ChIP-seq data for histone modification (Sharon et al. 2012 Cell) showed that decreased H3K4me3 and increased H3K27me3 of TSS of SCN5A during CM differentiation. Their data is not consistent with lamin A/C ChIP-qPCR data that the authors observed in control iPSC-CMs.

3. The authors observed abnormal sarcomeric structure in mutant iPSC-CMs. Did the authors check the expression of sarcomeric protein such as TNNT2, alpha-SMA or MYH6/7 in control and mutant iPSC-CMs?

--

Reviewer #3 (Remarks to the Author):

The authors have significantly improved the manuscript with additional data and analyses. These revisions address all the questions raised by the reviewers. I agree with the authors assessment that the data considering remodelin is best kept as starting point for future work and left out of this manuscript.

I have 1 remaining minor comment:

in Supplemental figure 10D the legend to the first two bars (red and black) is unclear/missing. I assume based on the data that these bars are a reproduction of the data in Suppl. Fig 6C. If this is indeed the case adjusting the legend (LMNA-K219T instead of LMNA) and a sentence explaining the bars is sufficient. If not the authors need to explain what these bars represent.

RESPONSE TO REFEREES

We thank again the editor and the referees for the positive feedback and for recognizing the efforts made to perform the new experiments included in the revised version of the manuscript.

The new comments and recommendations allowed us to further improve the quality of our work.

Below we reproduce in ***bold italics*** the referees' comments, followed by our response and how we have modified the manuscript when appropriate.

Replay referee #1

In the revised version of this manuscript authors performed a huge amount of additional experiments, improving the quality of the work.

I am satisfied of how they address my concerns, however I have still some comments on the representation and interpretation of new data.

We thank the reviewer for the positive feedback on the revised version of our manuscript and for the further suggestions, which we believe have provided us with hints for a better interpretation of the results and the molecular mechanism.

Figure 5A and Supplementary Figure 9:

I noticed that the ChIP experiments were represented differently, In Figure 5A as % of the input, in supplementary Figure 9 as fold change over the IgG. ChIP experiments should be comparable along the manuscript and preferably showed as % of input. In line with this observation, the binding measured at d0 in wt cells on SCN5A locus (Supplementary figure 9) should be similar to the one measured in Figure 5A. Why they are so different? Does data representation influence the result? Are they using different primer pairs (and if so, why?). Are iPSC conditions different? If so I would add the condition presented in Figure 5A in the Supplementary figure 9 graph. If the authors can confirm this different dynamic of Lamin A binding at the SCN5A locus this panel should be placed in the Figure 5, because it represents an important piece of information about the molecular mechanism.

As requested, we have re-analysed the ChIP experiments shown in Supplementary Figure 9C and represented data as % of input. Using this representation, the enrichment of Lamin A/C on *SCN5A* promoter at day 12 of cardiac induction in K219T cells was confirmed, while significance at day 8 was lost.

In order to further confirm this data, we have repeated the ChIP experiments using new sets of samples of differentiating CNTR and K219T-iPSC: data confirmed that differences between CNTR and K219T emerge at day 12 of differentiation. This result is also consistent with those from Western Blot experiments, showing the maximal accumulation of Lamin A/C protein at day 12 during cardiac differentiation (see Figure 1A of this reply to reviewers). Furthermore, a retrospective analysis of RNA sequencing data during differentiation of control iPSC lines (preliminary – unpublished) indicated that *SCN5A* gene expression highly increases in the window between d10 and 15 (Figure 1B show results from the validation RT-PCR).

Since we concur with reviewer #1 that this is an important piece of information, in the new version of the manuscript the ChIP against Lamin A/C during differentiation has been moved from Supplementary Figure 9 to Figure 5 (as panel C), as suggested. Text has been modified accordingly (page 11, line 13-16).

Regarding the amount of binding at d0, we would like to clarify that this cannot be similar to the one measured in the Figure 5A since the two analysed conditions are different: Figure 5A refers to highly differentiated CMs (cells maintained in culture for approximately 35-40 days in total), while

d0 in the differentiation experiment corresponds to undifferentiated iPSCs, that do not (or barely) express Lamin A/C (see Figure 1A of this reply to reviewers and Constantinescu et al., 2006). Thus, it is correct that, in this condition, we observed no (or very low) binding of Lamin A/C in samples from both CNTR and K219T. *SCN5A* expression in that condition is probably inhibited through an alternative mechanism (i.e. LBR).

Figure 1 - Expression of Lamin A/C protein and *SCN5A* gene during cardiac differentiation. A) Western Blot showing the maximal accumulation of Lamin A/C protein at day 12. As expected, very little expression of Lamin A/C is detectable in undifferentiated iPSC (d0). B) RT-PCR targeting *SCN5A* gene during cardiac differentiation, showing the induction of the gene is predominantly occurring between d10 and d15. Data are presented relative to undifferentiated iPSC (d0) and normalized for the expression of the HPRT housekeeping gene.

Figure 5D and E:

- In the revised version of the manuscript, authors, for the first time hypothesized a direct, physical interaction between mutated Lamin A and PRC2. I believe, also looking at new experiments that this is the right direction. However, in the text this is not clearly stated and FISH experiments showing a peripheral localization in K219T cells (Figure 5E) were not commented in the context of Lamin A-PRC2 physical interaction. If their hypothesis consider an aberrant and constitutive interaction between K219T Lamin with PRC2, it is possible to suppose that PRC2, already bound to the SCN5A locus is retained at the nuclear periphery, tethering the locus in a more repressive environment? Do I correctly interpret what they want to claim? In this case STED analysis should be repeated dividing nuclei in distinct zones (central and peripheral). If K219T-Lamin is retaining PRC2 in the nuclear periphery, analysis performed in the periphery zone should reveal a difference between wt and K219T-mutant more consistent than one performed in the central zone. In addition, quantitative analysis of SUZ12 immunofluorescence dividing nuclei in distinct zones (central and peripheral) could show a relocation of PRC2 in the K219T mutant.

The reviewer has interpreted our view in exact terms. The hypothesis we raised to explain the mechanism relies on a higher binding affinity between PRC2 and Lamin A/C in presence of the mutation. To further support this hypothesis and to satisfy the request from reviewer #2, we performed additional Co-IP experiments on control iPSCs overexpressing either wt or K219T-Lamin A/C. Results confirmed those arising from the imaging data and indicated a higher amount of PRC2 (detected with an antibody against Ezh2) bound to Lamin A/C in cells carrying the K219T mutation. These results have been included in the manuscript as Figure 7B. Text has been modified accordingly (page 13 line 1-5) and description of the methodology included in the Supplementary Methods (page 11 line 9-16).

Regarding the interpretation of 3D-FISH experiments results in the context of the PRC2-Lamin A/C

interaction, we agree with the reviewer: we take into proper account the hypothesis he/she raised as a potential mechanism acting in presence of the K219T mutation. It is in fact conceivable that a preferential localization of PRC2 together with Lamin A/C at the *SCN5A* locus exerts a commutative repressive action.

Following the reviewer's suggestion, we have reanalysed the STED data (z-stack images) subdividing the nucleus in three equal parts in length, one peripheral, one central and one in between (referred as middle zone in the manuscript). Results from these analyses have revealed that PRC2-Lamin A/C distribution in the nucleus varies between CNTR and K219T-CMs in all compartments, with a portion of the complex located significantly closer to the nuclear periphery in K219T-CMs compared to CNTR-CMs. These results have been included in the new Figure 7 (panel D), in the new Supplementary Figure 10, described in the text at page 13 (line 8-18) and discussed at pages 19 (line 9-24) and 20 (1-4).

We were unable to directly correlate these results to the position of *SCN5A* locus (from 3D-FISH experiments) since a STED experiment that assesses simultaneously *SCN5A*, Suz12 and Lamin A/C was not technically possible with the available tools. However, we can speculate that the more abundant fraction of PRC2 bound to Lamin A/C at the nuclear periphery in K219T-CMs would probably act at the *SCN5A* locus (among others) and result in the repression of its transcription.

Interestingly, as also discussed in the text (page 13, line 15-18), a portion of the complex is also enriched at the nuclear interior. This finding is consistent with previously published observations on cells lacking Lamin A/C (Marullo et al., 2016), demonstrating the causal relationship between Lamin A/C and the intra-nuclear localization of PcG proteins, and with the differential effect of *LMNA* mutations on the respective Lamin A/C protein dynamics (Serebryanny et al., 2018).

Detailed description of the strategy used for the analysis is provided in the Supplementary Methods (page 13 line 6-16).

- Representative images from wt and mutant are recommended.

In the new revised version of the manuscript we have included a representative image of both CNTR and K219T-LMNA CMs after 3D-map reconstruction (see Figure 7D).

We did not include a STED image (as in Figure 7C) from both conditions because no differences are appreciable by eye.

- Statistic analysis of graph in Figure 5D is not convincing. How many measurements they performed? How many independent clones? Which cells were used? Which statistic analysis was used?

Experimental details on the PRC2-LaminA colocalization have been expanded and are available in the Supplementary Methods (page 13, line 6-16) and in the relative figure legend. In brief, to answer the reviewer's question, we analysed n=16 cells in each condition (differentiated from 2 different clones per subject). For the statistical analysis, an unpaired Mann-Whitney test with 95% confidence interval was used.

- Authors wrote in the figure legend: "On the right, a representative image of a nucleus stained for the two proteins" but the image is on the left.

We apologize for the oversight and have changed it accordingly in the new version of the manuscript.

Overall, I suggest to re-analyse some data and to add few experiments to better support their working hypothesis. In addition, the proposed molecular mechanism should be described clearly along the text (abstract, result and discussion).

In this new revised version, we have performed all the analyses suggested by the reviewer;

furthermore, description of the proposed molecular mechanism has been extended throughout the manuscript in the appropriate paragraphs. Changes in the text are highlighted in blue.

Reply to Reviewer #2

Major comments

I appreciate the authors' efforts to address my previous comments, especially generation of additional iPSC lines (Fig. 7). The current revised manuscript showed increase quality and quantity of data that support below two main points of this manuscript. I have no additional questions for these main points:

- 1) Lamin A/C mutation (K219T) induces abnormal sodium current of mutant iPSC-CMs.*
- 2) Decreased SCN5A is required for disease phenotype in mutant iPSC-CMs.*

However, it is still difficult for me to be convinced that lamin A/C mutation decreases the expression of SCN5A gene in mutant iPSC-CMs through epigenetic modulation, which could be essential molecular mechanism throughout the entire manuscript. The main question is how lamin A/C mutation (K219T) affects the transcriptional regulation of SCN5A in mutant iPSC-CMs.

In this revised manuscript, the author showed that lamin A/C and Suz12 (PRC2 complex protein), is highly enriched in the promoter region of SCN5A gene (Fig. 5C). The authors also claimed that mutant lamin A/C protein has higher binding affinity with Suz12 (Fig. 5D). In this point, I am confused what the suggested mechanism for the abnormal expression of SCN5A gene in mutant iPSC-CMs is. Do these data mean that lamin A/C mutation induces global alteration of PRC2 composition and that abnormal PRC2 occupancy recruits lamin A/C into the promoter region of SCN5A?? Or Lamin A/C mutation results in new LADs including SCN5A region and additionally recruits PRC2 complex into the promoter region of SCN5A??

We appreciate the reviewer is satisfied with the functional data provided in the revised version and we do hope that the new results and analyses we are providing in the second revision will convince him/her also about the described molecular mechanism.

In brief, we do propose that in CMs carrying the K219T mutation, *SCN5A* is retained at the nuclear periphery where it is bound by Lamin A/C and PRC2, that, in presence of the mutation, show an increased binding affinity. This cooperative action of Lamin A/C and PRC2 results in the diminished expression of *SCN5A* gene and leads to the cell phenotype.

In support of this view, in this new version of the manuscript we have added further evidence on the physical interaction between Lamin A/C and PRC2 (Co-IP experiment in new Figure 7B) and on the differential dynamics of the Lamin A/C-PRC2 complex in CNTR and K219T-CMs (Figure 7D and Supplementary Figure 10). Whether the peripheral regulation described here occurs in a newly formed LAD or not is speculative and we cannot answer this question based on the available data.

However, as discussed in the previous reply, bioinformatics analyses conducted gathering data from cell types other than CMs (i.e., fibroblasts, HeLa cells, adipose stem cells) indicate that the *SCN5A* gene maps close to *SCN10A* and *SCN11A*, which are located in a LAD domain (Lund et al., 2013). As shown in Figure 5A, the analysis of the binding of Lamin A/C to *SCN10A* in CNTR and K219T-CMs revealed an enrichment in both conditions, regardless of the presence of the mutation, while a differential enrichment was detected on *SCN5A*. Based on these data, together with the differentiation-specific induction of *SCN5A* gene, we can speculate that this is probably located in the variable portion of the LAD (Remarks to this in the text can be found at page: 11 line: 13-19).

First, if authors want to claim that mutant lamin A/C has more binding affinity with Suz12, then additional experiment such as immuno-precipitation (IP) is needed to check the difference in the binding capacity between wild-type lamin A/C and mutant lamin A/C.

As requested, we performed Co-IP experiments. Results showed that K219T mutant LaminA/C possesses a higher binding capacity for PRC2 (Ezh2) compared to the CNTR protein. This additional evidence has been included in the new version of the manuscript in the new Figure 7 (as panel B).

Second, if authors want to claim that there is alteration of LADs in mutant iPSC-CMs compared to control iPSC-CMs, then global occupancy of lamin A/C on the genome should be analyzed. Because the mean size of LAD is 1-2MB (2016 Gesson et al. Genomic Research), there is limitation to identify the alteration of LADs using ChIP-qPCR.

We concur with the reviewer that determining the global occupancy of Lamin A/C on the entire genome would be of high interest and will unveil relevant mechanistic information on the molecular events driven by K219T mutation in LaminA/C. However, as we already discussed in our previous reply, these investigations fall outside the scope of this manuscript, and besides require a significant additional amount of time. It will however represent the inspiration for further work on this topic from our group.

Again, our investigation here has been focused on the understanding of the mechanism behind the regulation of *SCN5A* gene, starting from results of the functional electrophysiological screening. According to the mechanism we propose, the mutated Lamin A/C cooperates with PRC2 in downregulating *SCN5A*, leading to decreased sodium current density and slower conduction velocity. The hypothesis of an alteration of LADs in mutant CMs raised in the manuscript is speculative. However, we think the reviewer will agree with us that, similarly to other laminopathy models, rearrangement of LADs is likely to occur also in CMs carrying *LMNA* mutations.

Regardless of global distribution of lamin A/C, what is the possible mechanism that mutation (K219T) on the lamin A/C leads to abnormal occupancy of lamin A/C on the promoter region of SCN5A?

Based on the data on cardiac differentiation, we believe that in mutant cells *SCN5A* is retained to the nuclear periphery, while in the control cells this genomic region is released in a time-regulated manner and is essential for the proper functional maturation of the CMs. Whether the persistence of the *SCN5A* gene at the nuclear periphery is mediated by direct binding of mutant Lamin A/C to the gene's promoter or is mediated through interaction with other inner nuclear membrane proteins has to be determined.

Above all, these data suggest that the alteration of either LADs or binding affinity to PRC2 complex will result in global gene expression changes. Again, I highly speculate that not only SCN5A gene but also the combination of various genes should contribute to the disease phenotype of mutant iPSC-CMs. I think the authors need to clarify their hypothesis and reinforce it through additional results.

We agree with the reviewer on this point and we never excluded this possibility. Indeed, as also highlighted in the text (page: 20 line 5-9), we do expect a Lamin A/C-driven transcriptional modulation also on other genes and we have preliminary data supporting this assumption. As also mentioned in our first reply to the reviewer, preliminary RNA-sequencing data, not included in this manuscript, obtained from CNTR- and K219T-CMs indicated a variety of differentially modulated genes between the two cellular models that are involved in several processes relevant for the cardiac function other than cardiac conduction, such as contractility and cell metabolism. As discussed earlier in this reply, the reason why we focused on *SCN5A* gene lies in the results from the electrophysiology experiments indicating that the sodium current, and therefore $\text{Na}_v1.5$, was grossly altered in K219T-CMs. We therefore based our study on the hypothesis that alteration of *SCN5A*

gene expression could be a leading mechanism for the defects in cardiac conduction and excitability induced by K219T *LMNA* mutation.

Minor comments

1. To my knowledge, in hESCs (or iPSCs), the lamin B1 is highly expressed, but lamin A/C protein level is very low. How could the authors perform ChIP with native lamin A/C in D0 of CM differentiation (Supplementary Fig. 9C)? Did authors test the protein level of lamin A/C during CM differentiation and compare it in control and mutant lines?

The reviewer is correct: Lamin A/C is not (or barely) expressed in pluripotent stem cells (see Figure 1A of this reply to reviewers); this was used as negative control. In the new set of experiments included in the new Figure 5C, binding detected at d0 is, as expected, close to zero. The reason why the previous experiments showed some binding may depend on the presence of a subpopulation of more committed cells in the bulk of undifferentiated iPSCs; in fact, cardiac differentiation protocols require cells to reach 90-95% confluence at d0, a condition in which precocious and undesired differentiation may occur. In the new set of experiments we were more careful in preventing differentiation before harvesting cells at d0.

Regarding the levels of Lamin A/C protein during differentiation, we have not made a comparison between the two conditions. However, no differences were appreciable from Lamin A/C immunofluorescence experiments conducted on CNTR and K219T differentiated CMs.

2. Previous ChIP-seq data for histone modification (Sharon et al. 2012 Cell) showed that decreased H3K4me3 and increased H3K27me3 of TSS of SCN5A during CM differentiation. Their data is not consistent with lamin A/C ChIP-qPCR data that the authors observed in control iPSC-CMs.

Probably the reviewer refers to (Paige et al., 2012) – doi 10.1016/j.ymeth.2018.08.009); if this is the case, we wish to report below our considerations on the ChIP-seq data included in that manuscript, analysing the parameters highlighted by the reviewer.

When limiting analysis to the TSS (-2Kb/+2 Kb), as stated by the reviewer, we did not detect any significant enrichment/depletion of H3K27me3 and H3K4me3 throughout differentiation. However, we agree that if we consider the entire gene body, there is a slight increase of H3K27me3, while H3K4me3 does not significantly change. This data correlates with the gene expression profile (Affimetrix data) shown by the authors, indicating a significant decrease of the expression of *SCN5A* gene. In contrast, since *SCN5A* gene is a specific marker of cardiomyocytes, its expression must increase during differentiation and be highly enriched in highly purified differentiated CMs. An analysis of another marker of CMs, *CACNIAC* (a calcium channel), in the same data set revealed a similar trend of expression. Thus, it is more than likely that the data presented in Paige et al. are biased for contaminant cells (non-cardiomyocytes), particularly, but not only, fibroblasts. As a matter of fact, an analysis of vimentin expression confirms this hypothesis, since a raise of vimentin expression in differentiating cells is concomitant with a complete loss of H3K27me3 by d14. This result could depend upon the differentiation protocol used in this study, which still relies on EB aggregation; in the last 6 years, protocols to differentiate CMs have improved to the point that it is possible to obtain a population made up of 90% CMs, as is the case for the present study. In summary, the aforementioned histone modification data could be confusing if the *SCN5A* gene is taken into account, since the tissue culture conditions are not selective for CM differentiation.

Since our analysis refers to terminally differentiated CMs (10 days of induction + additional 25-30 days of maturation), the two datasets cannot be compared.

Finally, in line with our observation, data from differentiating mouse ESCs (Wamstad et al., 2012), which represent a more stable system for inducing CMs, show an increase of *SCN5A* expression (>

20-fold), accompanied by a 2.4-fold increase of the H3K4me3/H3K27me3 ratio in the last two stages of differentiation (CP: cardiac precursor, CM: cardiomyocyte).

3. The authors observed abnormal sarcomeric structure in mutant iPSC-CMs. Did the authors check the expression of sarcomeric protein such as TNNT2, alpha-SMA or MYH6/7 in control and mutant iPSC-CMs?

For this work, we only analysed alpha-sarcomeric actinin expression by immunofluorescence in the experiments aimed to assess sarcomeric organization. From this data, no remarkable differences in the levels of expression of the protein were detectable between CNTR and K219T-CMs.

--

Reply to Reviewer #3

The authors have significantly improved the manuscript with additional data and analyses. These revisions address all the questions raised by the reviewers. I agree with the authors assessment that the data considering remodelin is best kept as starting point for future work and left out of this manuscript.

***I have 1 remaining minor comment:
in Supplemental figure 10D the legend to the first two bars (red and black) is unclear/missing. I assume based on the data that these bars are a reproduction of the data in Suppl. Fig 6C. If this is indeed the case adjusting the legend (LMNA-K219T instead of LMNA) and a sentence explaining the bars is sufficient. If not the authors need to explain what these bars represent.***

We are grateful to the reviewer for the appreciation of our work and we are glad he/she is satisfied with it.

To reply to the last minor comment, we confirm that the first two bars in Supplementary Figure 10D (which is now Supplementary Figure 11D) reproduce data of Figure 2B) and refer to CNTR (black) and LMNA-K219T (red) CMs. The same is applicable to Supplementary Figure 6C. In the new revised version, we modified the figure legends accordingly.

Cited references

- CONSTANTINESCU, D., GRAY, H. L., SAMMAK, P. J., SCHATTEN, G. P. & CSOKA, A. B. 2006. Lamin A/C expression is a marker of mouse and human embryonic stem cell differentiation. *Stem Cells*, 24, 177-85.
- LUND, E., OLDENBURG, A. R., DELBARRE, E., FREBERG, C. T., DUBAND-GOULET, I., ESKELAND, R., BUENDIA, B. & COLLAS, P. 2013. Lamin A/C-promoter interactions specify chromatin state-dependent transcription outcomes. *Genome Res*, 23, 1580-9.
- MARULLO, F., CESARINI, E., ANTONELLI, L., GREGORETTI, F., OLIVA, G. & LANZUOLO, C. 2016. Nucleoplasmic Lamin A/C and Polycomb group of proteins: An evolutionarily conserved interplay. *Nucleus*, 7, 103-11.
- PAIGE, S. L., THOMAS, S., STOICK-COOPER, C. L., WANG, H., MAVES, L., SANDSTROM, R., PABON, L., REINECKE, H., PRATT, G., KELLER, G., MOON, R. T., STAMATOYANNOPOULOS, J. & MURRY, C. E. 2012. A temporal chromatin signature in human embryonic stem cells identifies regulators of cardiac development. *Cell*, 151, 221-32.
- SEREBRYANNYY, L. A., BALL, D. A., KARPOVA, T. S. & MISTELI, T. 2018. Single molecule analysis of lamin dynamics. *Methods*.
- WAMSTAD, J. A., ALEXANDER, J. M., TRUTY, R. M., SHRIKUMAR, A., LI, F., EILERTSON, K. E., DING, H., WYLIE, J. N., PICO, A. R., CAPRA, J. A., ERWIN, G., KATTMAN, S. J., KELLER, G. M., SRIVASTAVA, D., LEVINE, S. S., POLLARD, K. S., HOLLOWAY, A. K., BOYER, L. A. & BRUNEAU, B. G. 2012. Dynamic and coordinated epigenetic regulation of developmental transitions in the cardiac lineage. *Cell*, 151, 206-20.

Reviewers' comments:

Reviewer #1 (Remarks to the Author):

The authors addresses the comments I made in a satisfactory manner.
The text needs an english revision.

Reviewer #2 (Remarks to the Author):

I appreciate the authors' efforts to address previous comments. I agree with author's claim that the identification of genome-wide distribution of LMNA in WT and K219T is out of scope of current manuscript. However, I think the authors need to consolidate the mechanism part to make sure their hypothesis is more robust instead of flimsy. Here below I provided few more comments:

1) The authors newly performed IP analysis to show that K219T Lamin A/C has more binding affinity with Ezh2 (Fig. 7B) and claimed that this could be possible mechanism for the abnormal suppression of SCN5A. However, I think the authors need to provide more quantitative data for the IP analysis. Although the authors claimed that K219T binds more to Ezh2, it is difficult for me to see a big difference in the IP column. For example, there are bright spots in Ezh2 row that disrupt real signal from bands.

To my knowledge, Lamin A/C and PRC2 complex are highly expressed in CM, so I highly recommend to perform co-IP with native LMNA or PRC2 in CMs.

2) The authors are claiming "Lamin A/C-PRC2 interplay" as a main mechanism of abnormal expression of SCN5A in K219T. However, the authors should include additional supporting data that explain why PRC2 complex is selected for potential target of mutant lamin A/C. There are many possible mechanisms for the epigenetic changes on SCN5A promoter in K219T Lamin A/C, which the authors observed in Fig 5 and Supplementary Fig 9.

Because PRC2 and LMNA have distinct enrichment pattern of genomic distribution (LMNA covers 1~2 MB whereas PRC2 restricted within 1 kb), more data is needed to prove that LMNA and PRC interplay in the regulation of specific gene such as SCN5A. For example, authors should check the enrichment of PRC complex on the genes that share LAD with SCN5A.

3) Again, if "Lamin A/C-PRC2 interplay" is key mechanism of K219T-CMs, it will affect the global epigenetic status of the entire genome. Hence the authors need to explain or show evidences why SCN5A is mainly associated with disease phenotype among other target genes

RESPONSE TO REFEREES

We thank again the editor and the referees for the positive feedback and for recognizing the efforts made to address the comments they raised. Nonetheless, reviewer #2 still has some questions that we hope will be satisfied by the following response.

Below we reproduce in ***bold italics*** the referees' comments, followed by our response and how we have modified the manuscript when appropriate.

Reply to Reviewer #1

The authors addressed the comments I made in a satisfactory manner.

The text needs an english revision.

We thank the reviewer for the appreciation of our work and we are glad he/she is satisfied with it. As suggested, text has been proofread.

Reply to Reviewer #2

I appreciate the authors' efforts to address previous comments. I agree with author's claim that the identification of genome-wide distribution of LMNA in WT and K219T is out of scope of current manuscript. However, I think the authors need to consolidate the mechanism part to make sure their hypothesis is more robust instead of flimsy. Here below I provided few more comments:

1) The authors newly performed IP analysis to show that K219T Lamin A/C has more binding affinity with Ezh2 (Fig. 7B) and claimed that this could be possible mechanism for the abnormal suppression of SCN5A. However, I think the authors need to provide more quantitative data for the IP analysis. Although the authors claimed that K219T binds more to Ezh2, it is difficult for me to see a big difference in the IP column. For example, there are bright spots in Ezh2 row that disrupt real signal from bands.

To my knowledge, Lamin A/C and PRC2 complex are highly expressed in CM, so I highly recommend to perform co-IP with native LMNA or PRC2 in CMs.

2) The authors are claiming "Lamin A/C-PRC2 interplay" as a main mechanism of abnormal expression of SCN5A in K219T. However, the authors should include additional supporting data that explain why PRC2 complex is selected for potential target of mutant lamin A/C. There are many possible mechanisms for the epigenetic changes on SCN5A promoter in K219T Lamin A/C, which the authors observed in Fig 5 and Supplementary Fig 9. Because PRC2 and LMNA have distinct enrichment pattern of genomic distribution (LMNA covers 1~2 MB whereas PRC2 restricted within 1 kb), more data is needed to prove that LMNA and PRC interplay in the regulation of specific gene such as SCN5A. For example, authors should check the enrichment of PRC complex on the genes that share LAD with SCN5A.

3) Again, if "Lamin A/C-PRC2 interplay" is key mechanism of K219T-CMs, it will affect the global epigenetic status of the entire genome. Hence the authors need to explain or show evidences why SCN5A is mainly associated with disease phenotype among other target genes

We are glad the reviewer is now concurring with us that the genome-wide analysis of Lamin A/C distribution is beyond the scope of this manuscript.

However, he/she still raised some criticisms on the solidity of the described molecular mechanism. Before going into the discussion of the specific requests, we would like to highlight that in the previous two rounds of revision we generated an enormous amount of additional data, including experiments on the isogenic lines and the co-IP, as the reviewer specifically requested. The mechanism was therefore consolidated using different approaches. In addition, the correspondence between the molecular results and the functional data further strengthens the value of our findings. Thus, from our point of view, the mechanism we are describing is anything but weak.

However, we understand the reviewer's attempts to be as meticulous as possible to improve further the quality of the data presented and to dissect the mechanism more deeply. In light of this consideration, we carefully analysed the concerns he/she raised and made an effort to address or discuss them, in consideration of the time constrains and of the actual advancement for the manuscript.

The answers to the specific requests/criticisms are listed below as point-to-point rebuttal.

- 1) In order to satisfy the reviewer's request, we performed the Co-IP experiment on CMs differentiated from our iPSC lines. The obtained results are in line with those gathered from super-resolution microscopy and confirm that K219T mutant Lamin A/C possesses a higher binding capacity for PRC2 (Ezh2) compared to the CNTR protein. Results have been added

in the new version of the manuscript in the panel B of Figure 7.

We would nonetheless like to make a consideration on this piece of data. Based on the distribution profiles of Lamin A/C, PRC2 and their complex in CM nuclei, we would not expect a big difference in the binding of the two proteins in the two experimental conditions at the global level. As also highlighted by the reviewer, Lamin A/C and PRC2 are highly expressed in CMs, and are important players in regulating gene expression and cell fate. Thus, dramatic changes in their bound proportions would probably lead to extremely severe cell phenotypes and to an impairment in cell differentiation, events we did not observe.

- 2) We thank the reviewer for the effort in trying to deepen the molecular mechanism regulating the interplay between mutant Lamin A/C and PRC2. However, we believe that the suggested experiment would not add any additional value to our work since, regardless of the emerging result, the proposed mechanism of Lamin A/C-PRC2 interplay in regulating the *SCN5A* gene would still be valid for several reasons. First, PRC2-Lamin A/C interplay might selectively target specific genes, depending also on the influence of other epigenetic factors and of chromatin conformation, that cannot be assessed by a simple ChIP-PCR; second, the two genes sharing the LAD domain with *SCN5A* (namely, *SCN10A* and *SCN11A*) are not expressed in the CMs, neither they are modulated between the two conditions, but rather they are neuronal isoforms of the sodium channel $Na_v1.8$ and $Na_v1.9$ (Verkek A.O. et al, 2012, Dib-Haji S.D. et al., 2015, Marban E. et al, 1998 and unpublished RNA-seq data by us - see in Figure 1 of this reply to reviewers), so that any ChIP-PCR result on those genes in CMs would be of difficult interpretation and potentially confounding; third, in the manuscript we provided evidence that Lamin A/C-PRC2 interplay is not exclusive in regulating *SCN5A* gene expression, rather binding of Lamin A/C itself and gene positioning play also a role. In addition, we never excluded the possibility (actually we have hypothesized it) that other layers of regulations, such as other chromatin binding factors and long-range chromatin interactions, may be involved (Gonzalez-Sandoval A. & Gasser, S.M. 2016). In the manuscript, we decided to further investigate the potential role of PRC2 because of the results on the enrichment of H3K27me3 marker at the TSS of *SCN5A* and of previous reports showing that Lamin A/C interacts with PRC2 in other cell types (Cesarini E. et al., 2015, Marullo F. et al., 2016). We agree with the reviewer that investigating the effect of PRC2-Lamin A/C interaction on different genes is an area of interest, but this should be done either on genes that are expressed and modulated or through genome-wide approaches and conformational studies that, as already agreed, are beyond the scope of this manuscript.

Figure 1 – RNA sequencing results for selected sodium channels’ encoding genes in CNTR and LMNA-CMs. A) Differential expression analysis between CNTR- and LMNA-CMs for SCN10A and SCN11A genes, showing no expression is detectable in both conditions from both genes (RPKM < 0.5). A gene is considered expressed if RPKM ≥ 1 in at least one condition. Differential expression analysis for SCN5A gene has been included as control. B) Screenshot of IGV profiles of RNA-seq coverage on the selected loci (SCN10A, SCN11A and SCN5A). Data are scaled on threshold of expression (-1,0: reverse strand).

- The reviewer already raised this point to our attention previously in the review process and is difficult for us to understand what is still not clear. As already extensively discussed both in the text and in the reply to the reviewer, we never excluded the possibility that other genes may be modulated by global epigenetic changes in LMNA mutant cells and influence the cell phenotype; we have actually hypothesized the contrary. The reasons why we focused on SCN5A are based on the results from the electrophysiological studies in our models and on

the main role of SCN5A, among the other potential target genes, on cardiac conduction. Indeed, we generated extensive electrophysiological data demonstrating that Na⁺ current is profoundly affected in this type of cardiomyopathy. This gene plays a fundamental role in the regulation of cardiomyocyte action potential, being involved in many hereditary arrhythmias, including Long QT and Brugada syndromes. A brief reminder: in laminopathies with myocardial involvement, cardiac arrhythmias are the major cause of death. Our manuscript focuses on the effects of Lamin A/C mutation on the CM action potential, not on other pathophysiological aspects related to the disease (such as cytoskeletal and nuclear structural disorganization), which will be accounted for in other reports.

So, in brief, that SCN5A is mainly associated with the conduction defect is intrinsically related to the major role of its encoded protein, Nav1.5, in cardiac conduction, and this also clearly emerges from the functional analyses, in which the maximal upstroke velocity stands out as the most defective electrophysiological parameter in mutant cells. Furthermore, the major role of Nav1.5 also comes out from the impulse propagation experiments in which Nav1.5 is specifically inhibited by TTX: this showed that there is no residual conduction in CMs in which Nav1.5 is inhibited, supporting that the phenotype is mainly driven by the presence of the channel.

We hope the reviewer will be convinced now of the pivotal role of this gene in the phenotype we are investigating and concur with us on the choice to focus on its regulation.

Cited references

VERKERK, AO, REMME, C.A., SCHUMAKER, C.A., SCICLUNA, B.P., WOLSWINKEL, R., DE JONGE, B., BEZZINA, C.R., VELDKAMP, M.K. 2012. Functional NaV1.8 Channels in Intracardiac Neurons: The Link Between *SCN10A* and Cardiac Electrophysiology. *Circ Res*, 111, 333-43.

DIB-HAJI, S.D., BLACK, J.A., WAXMAN, S.G. 2015. Na1.9: a sodium channel linked to human pain. *Nat Rev Neurosci*, 16, 511-9.

MARBAN, E., YAMAGISHI, T., TOMASELLI, G.F. 1998. Structure and function of voltage-gated sodium channels. *J Physiol*, 508, 647-57.

GONZALEZ-SANDOVAL, A. & GASSER, S. M. 2016. On TADs and LADs: Spatial Control Over Gene Expression. *Trends Genet*, 32, 485-495.

CESARINI, E., MOZZETTA, C., MARULLO, F., GREGORETTI, F., GARGIULO, A., COLUMBARO, M., CORTESI, A., ANTONELLI, L., DI PELINO, S., SQUARZONI, S., PALACIOS, D., ZIPPO, A., BODEGA, B., OLIVA, G. & LANZUOLO, C. 2015. Lamin A/C sustains PcG protein architecture, maintaining transcriptional repression at target genes. *J Cell Biol*, 211, 533-51.

MARULLO, F., CESARINI, E., ANTONELLI, L., GREGORETTI, F., OLIVA, G., LANZUOLO, C. 2016. Nucleoplasmic Lamin A/C and Polycomb group of proteins: An evolutionarily conserved interplay. *Nucleus*, 7, 103-11.

REVIEWERS' COMMENTS:

Reviewer #2 (Remarks to the Author):

I appreciate the authors' efforts to address my previous comments. Although the authors provided rebuttal to my comments, my comments remain similar. It is still unclear to me how did the authors identify SCN5A as key factor for LMNA-DCM phenotype.

The author described "To prove this, we first analysed the expression of Nav1.5 encoding gene, SCN5A, through 22 RT-PCR, finding that the level of its transcript in K219T-CMs was significantly lower than in 23 control cells (Fig. 4A)" in main text. In addition, the author also described in rebuttal letter that abnormal Na⁺ current from EP results is main reason why they tested SCN5A. I looked through at the EP data and found ask the authors to at least look at the following questions.

1) As the authors state "Mutations of the LMNA gene, encoding the nuclear proteins Lamin A/C, cause dilated cardiomyopathy, and typically associate with conduction defects and arrhythmias." A manuscript attempting to unravel a delicate pro-arrhythmic mechanism for an abstract syndrome such as LMNA will need to base its findings/conclusions on results acquired from symptomatic patients and accordingly patient-specific iPSC-CMs displaying a pro-arrhythmic phenotype.

(i) In order to study a pro-arrhythmic mechanism, it is important for the authors to show that they have indeed recruited for the study symptomatic LMNA patients. The authors report that while LMNA patient 1 does NOT present a history of arrhythmia, interrogated LMNA patients 2 and 3 do indeed present a clinical history of arrhythmia (Supplementary Table 1). Currently, the only data presented in the manuscript for the interrogated patients is a "Y" (Yes) or "N" (No) for arrhythmia history.

Please provide patient cardiac ECG and parameters and any other data that will account for conduction abnormalities, in order for this pro-arrhythmic phenotype to be assessed at the patient level and accordingly associated with the functional and mechanistic phenotype at the cellular level.

(ii) To claim SCN5A as the mechanism, it is crucial for the authors to show that the acquired reprogrammed LMNA patient-specific iPSC-CMs naturally display a functional pro-arrhythmic AP phenotype. The authors do not show whether the acquired LMNA-iPSC-CMs display a pro-arrhythmic nature (e.g., EADs, DADs, triggered activity, irregular beating intervals, etc).

Please display a sequence of APs traces for each symptomatic patient derived line displaying the pro-arrhythmic nature of the cells. Please also include the percentage of pro-arrhythmic cells that are detected per patient line.

2) Currently (with a few exceptions) most of the data presented in all of the figures, of the main manuscript and the supplementary, present results for only one control line vs. one patient line. To claim SCN5A as the key factor, it is very important that the authors present (in the supplementary) data (both parameters and traces) for both control lines and all 3 patient lines. Please also similarly include data for all LMNA patient lines clones and genome-edited corrected line clones.

3) Numerous papers have shown that iPSC-CMs present 3 different action potential subtypes: ventricular-like, atrial-like and pacemaker-like. Each subtype is characterized by a different phenotype presenting different characteristics. Furthermore, disease phenotype many times is only displayed in specific subtype cells (for example only in ventricular or in both ventricular and atrial but not in pacemaker), shading more important light on disease mechanism. This is why when presenting action potential parameters and APs tracing, it will be very important for the authors to identify, separately characterize, and individually present each subgroup. Worryingly, the authors currently present data analysis for all the different AP subtypes as one joint group.

The importance of this request is very apparent in Figure 1. In Figure 1B (left panel), it is notable that a significant number of K219T cells present a depolarized maximal diastolic potential (MDP) in comparison to the control cells. These depolarized cells may very well be pacemaker-like cells (which

present a depolarized MDP in comparison to ventricular- and atrial-like cells). Meaning that the acquired K219T recordings have by chance a larger population of pacemaker cells in comparison to the control group. This would of course than also affect other parameter calculations for the K219T cells, such as AP amplitude, overshoot, and maximal upstroke velocity. Therefore, deriving conclusion from a mixed population of iPSC-CMs subtypes may lead to conclusion that will not necessarily reflect disease phenotype but rather iPSC-CMs subcellular population phenotype.

Please, reorganize your data to present ventricular, atrial, and pacemaker data (both parameters and tracings) individually (please make sure to include n=x for each group). Please also describe and explain any difference in disease phenotype present in the 3 different sub groups. This will be important to the overall understanding of the mechanism presented by this manuscript.

4) The authors go to great length to correct the K219T mutation using CRISPR/Cas9-mediated genome editing and test to see how the correction effects SCN5A gene expression and sodium current density. Yet surprisingly, no functional AP data at the single cell level (Figure 1) nor conduction data at the multicellular level (Figure 3) is presented for the corrected line.

To confirm the authors' conclusions, it will be necessary to test the direct effects of the correction on conduction velocity, AP parameters, and pro-arrhythmic activity in the genome-edited corrected line. This data should than be accordingly compared to data acquired from both control and K219T lines.

5) The authors described in rebuttal letter that "second, the two genes sharing the LAD domain with SCN5A (namely, SCN10A and SCN11A) are not expressed in the CMs, neither they are modulated between the two conditions, but rather they are neuronal isoforms of the sodium channel Nav1.8 and Nav1.9 (Verkek A.O. et al, 2012, Dib-Haji S.D. et al., 2015, Marban E. et al, 1998 and unpublished RNA-seq data by us - see in Figure 1 of this reply to reviewers)".

How did authors check SCN5A, SCN10A and SCN11A are located in same LAD? Did authors utilize published data set for LMNA (or LMNB1 ChIP-seq)? In LMNA-related research, most researchers have tested the concept of "LAD (Lamin associated domain)" to explain their hypothesis (J Cell Biol. 2015 Jan 5;208(1):33-52) (J Cell Biol. 2017 Sep 4;216(9):2731-2743) (Hum Mol Genet. 2018 Apr 15;27(8):1447-1459). And a recent paper showed how LAD correlated with H3K9me2 histone modification (Cell. 2017 Oct 19;171(3):573-587) that is closely related with PRC2 and H3K27me3 histone maker. I think the authors really have to seriously consider and reconcile their findings and hypothesis with these other papers.

RESPONSE TO REFEREES

We thank the editor and the reviewers for the general appreciation demonstrated for our work through the four rounds of revision of the manuscript. This long process certainly helped us in improving data quality, strengthen our hypotheses and conclusions.

Reviewer #2 however requested additional explanation and new supporting data. Below in ***italics*** are the referees' comments, followed by our response and how we have modified the manuscript when appropriate.

Reply to Reviewer #2

I appreciate the authors' efforts to address my previous comments. Although the authors provided rebuttal to my comments, my comments remain similar. It is still unclear to me how did the authors identify SCN5A as key factor for LMNADCM phenotype. The author described "To prove this, we first analysed the expression of Nav1.5 encoding gene, SCN5A, through 22 RTPCR, finding that the level of its transcript in K219TCMs was significantly lower than in 23 control cells (Fig. 4A)" in main text. In addition, the author also described in rebuttal letter that abnormal Na⁺ current form EP results is main reason why they tested SCN5A. I looked through at the EP data and found ask the authors to at least look at the following questions.

We are glad the reviewer acknowledged our efforts to address the criticisms he/she raised in the last three rounds of revisions and we believe that, from a simple comparison between the first submitted version and the current one, it can be appreciated that this required an enormous amount of additional work and also the set-up of additional technologies in the lab.

It is unfortunate that after 4 revisions the methodology used by us is still unclear. Defects in Na⁺ currents were found using an unbiased approach by studying the electrophysiological properties of LMNA-CMP. Gene expression analysis and following *SCN5A* epigenetic dysregulation followed EP results.

The link between the reduction of dV/dT_{max} , the decreased I_{Na} currents and the choice to investigate *SCN5A* is apparent and has been referenced in the main text (page: 6, line:13-17) and extensively discussed in the previous replies to the reviewer.

To briefly sum up our work, we started from a comprehensive analysis of the action potential properties of CMs carrying the K219T mutation and identified dV/dT_{max} as the major altered parameter compared to control CMs. dV/dT_{max} , together with its related AP properties (overshoot and action potential amplitude) reflects the fast Na⁺ currents during the upstroke. Those Na⁺ currents depend on the functional expression of the sarcolemmal fast voltage-dependent Na⁺ channel, Nav1.5, which is encoded by the *SCN5A* gene.

We considered again all the new criticisms raised by the reviewer. What follows is an extensive reasoning on the criticisms raised by the reviewer.

As the authors state "Mutations of the LMNA gene, encoding the nuclear proteins Lamin A/C, cause dilated cardiomyopathy, and typically associate with conduction defects and arrhythmias." A manuscript attempting to unravel a delicate pro-arrhythmic mechanism for an abstract syndrome such as LMNA will need to base its findings/conclusions on results acquired from symptomatic patients and accordingly patient-specific iPSC-CMs displaying a pro-arrhythmic phenotype.

(i) In order to study a pro-arrhythmic mechanism, it is important for the authors to show that they have indeed recruited for the study symptomatic LMNA patients. The authors report that while LMNA patient 1 does NOT present a history of arrhythmia, interrogated LMNA patients 2 and 3 do indeed present a clinical history of arrhythmia (Supplementary Table 1). Currently, the only data presented in the manuscript for the interrogated patients is a "Y" (Yes) or "N" (No) for

arrhythmia history. Please provide patient cardiac ECG and parameters and any other data that will account for conduction abnormalities, in order for this proarrhythmic phenotype to be assessed at the patient level and accordingly associated with the functional and mechanistic phenotype at the cellular level.

As the reviewer correctly remarked, LMNA-CMP manifests with an extremely heterogeneous phenotype, without obvious genotype-to-phenotype correlations. Although the main Lamin A/C-associated phenotypes are dilated CMP and conduction system diseases (CSDs), the disease displays variable expressions and age-dependent penetrance, so that patients may show only CMP or CSDs or still no phenotype, despite carrying the same mutation. The CSDs also present with very heterogeneous manifestations, ranging from atrio-ventricular block to sino-atrial dysfunctions, atrial fibrillation, symptomatic bradyarrhythmias, supraventricular/ventricular arrhythmias, and many other forms of arrhythmic manifestations. The presence of such conduction disturbances represents a risk for developing fatal arrhythmias. So, based on this extreme heterogeneity of the disease, we do not believe that the enrolled patients should necessarily be symptomatic.

However, in order to satisfy the requests from both the reviewer and the editor, in this new submitted version we have implemented the clinical data of the selected patients and added patients' representative ECGs (new Supplementary Figure 1) and more clinical information in Supplementary Table 1, especially in relation to conduction abnormalities, such as an electrophysiological study (EPS), echocardiography and relevant clinical events, such as pacemaker (PM) and implantable cardioverter defibrillator (ICD) implantation and history of onset of spontaneous arrhythmias. Specifically, the added clinical data indicate for patients LMNA #2 and LMNA #3 a positive history of spontaneous arrhythmias of different kinds (Y in the previous Supplementary Table 1, now better specified), while patient LMNA #1 did not refer any at the moment. However, we believe that the very young age of this patient compared to the age at diagnosis of the other two (42 and 39 years old for LMNA #2 and LMNA #3, respectively), might be the reason why the phenotype has not manifested yet.

In summary, the complexity of this pathology is at the basis of the heterogeneity of conduction abnormalities and clinical events. We believe that our study supports the notion that the K219T mutation leads to conduction disturbances, and is therefore arrhythmogenic, regardless of the clinical manifestations annotated in the patients.

(ii) To claim SCN5A as the mechanism, it is crucial for the authors to show that the acquired reprogrammed LMNA patient-specific iPSC-CMs naturally display a functional pro-arrhythmic AP phenotype. The authors do not show whether the acquired LMNA iPSC-CMs display a proarrhythmic nature (e.g., EADs, DADs, triggered activity, irregular beating intervals, etc). Please display a sequence of APs traces for each symptomatic patient derived line displaying the pro-arrhythmic nature of the cells. Please also include the percentage of pro-arrhythmic cells that are detected per patient line.

Response to this reviewer's criticisms is, at least to some extent, given by the results in Figure 3, where an increased susceptibility of mutant CMs to increased electrical stimulations and a decrease of their conduction velocity in the multi-cellular setting are shown. Disturbances in the conduction of the electrical impulse represent *per se* a substrate for the development of arrhythmic events. Performing the experiments the reviewer is requesting now, which were never requested in the previous revisions, would entail at least another full four months of work. In addition, and more importantly, results that may arise from the requested experiments, even if interesting, would not add any ground-breaking information to what is already included in the manuscript.

2) Currently (with a few exceptions) most of the data presented in all of the figures, of the main manuscript and the supplementary, present results for only one control line vs. one patient line.

To claim SCN5A as the key factor, it is very important that the authors present (in the supplementary) data (both parameters and traces) for both control lines and all 3 patient lines. Please also similarly include data for all LMNA patient lines clones and genome-edited corrected line clones.

We have to underline that the same criticism was raised by the reviewer in the first revision; we have already answered his/her question in a previous reply to reviewers and manuscript versions. Curiously, after two revisions, this criticism has emerged without taking into account that explanations have already been given and, apparently, accepted by the reviewer.

As already stated in the previous reply, all the experiments have been performed on CMs differentiated from two iPSC clones from each subject (2 CNTR and 3 LMNA) and data represented as a mean of those. This has been specified in the Material and Methods section (page: 20, lines: 2-3) and in the legends of the respective figures.

In addition to this, a separate graph providing the single values of the maximal diastolic potential (MDP) measured in the individual lines is included in the Supplementary Information, now as Supplementary Figure 5. As already discussed in the previous reply to the reviewer, we chose to show data from individual CMs lines for this specific parameter because MDP represents the electrophysiological recording from which all the other AP parameters were measured. In addition to this, all the parameters and traces of CMs from the independent clones have been now included also in the DATA SOURCES file.

3) Numerous papers have shown that iPSC-CMs present 3 different action potential subtypes: ventricular-like, atrial-like and pacemaker-like. Each subtype is characterized by a different phenotype presenting different characteristics. Furthermore, disease phenotype many times is only displayed in specific subtype cells (for example only in ventricular or in both ventricular and atrial but not in pacemaker), shading more important light on disease mechanism. This is why when presenting action potential parameters and APs tracing, it will be very important for the authors to identify, separately characterize, and individually present each subgroup.

Worrisomely, the authors currently present data analysis for all the different AP subtypes as one joint group. The importance of this request is very apparent in Figure 1. In Figure 1B (left panel), it is notable that a significant number of K219T cells present a depolarized maximal diastolic potential (MDP) in comparison to the control cells. These depolarized cells may very well be pacemaker-like cells (which present a depolarized MDP in comparison to ventricular and atrial-like cells). Meaning that the acquired K219T recordings have by chance a larger population of pacemaker cells in comparison to the control group. This would of course than also affect other parameter calculations for the K219T cells, such as AP amplitude, overshoot, and maximal upstroke velocity. Therefore, deriving conclusion from a mixed population of iPSCCMs subtypes may lead to conclusion that will not necessarily reflect disease phenotype but rather iPSCCMs subcellular population phenotype.

Please, reorganize your data to present ventricular, atrial, and pacemaker data (both parameters and tracings) individually (please make sure to include n=x for each group).

Please also describe and explain any difference in disease phenotype present in the 3 different sub groups. This will be important to the overall understanding of the mechanism presented by this manuscript.

Again, we are quite surprised by reviewer's request because analyses that distinguish recordings from working-like CNTR- and K219T-CMs from pacemaker-like cells, based on their AP characteristics, had been already performed and relative data discussed both in the main text and in the supplementary material since the very first submission.

In the last version of the manuscript, page 7, line 14-24, we wrote: *“Those data were confirmed when AP parameters were recorded in working-like CMs, which were selected on the basis of their AP*

characteristics using criteria we have previously described²². These additional evaluations were included in the first set of analyses to exclude potential underestimation of I_{Na} due to contamination with pacemaker cells, which are more depolarized and devoid of $Na_v1.5$. Indeed, differentiation of iPSCs into CMs is known to give rise to a mixed population of atrial-, ventricular- and pacemaker-like cells^{22,39}: these different cell populations are typified by distinct AP parameters that reflect on AP morphology and determine specific functional outcomes^{40,41}. In line with the results previously obtained in spontaneously active iPSC-CMs and in those electrically adjusted to -82mV, AP parameters recorded in working-like K219T-CMs were significantly decreased when compared with those measured in CNTR-CMs (Supplementary Table 3), with a decrement in dV/dt_{max} of 29.8% (Supplementary Fig. 5)".

Also, in that manuscript on page 16, line 16-29, we stated: "Importantly, the decreased cellular excitability seen in LMNA-CMs is coherent with the different AP characteristics and the biophysical parameters of all analysed cell populations (i.e., spontaneously active, electrically adjusted and working-like iPSC-CMs), further validating the use of this cellular model to study $Na_v1.5$ current and conduction in LMNA-CMP".

However, in order to satisfy the reviewer's request, in this new version we have included values and properties of the pacemaker-like cells (see Supplementary Table 3) and provided examples of traces in the DATA SOURCES file.

We are aware that now there are protocols available that allow differentiation of specific cardiac cell types (atrial, ventricular and sino-atrial node cells), but they were published in 2017, the same year in which our manuscript was first submitted (August 3, 2017). Furthermore, those protocols are quite complex and would have needed extensive setting-up, and this is the main reason why we did not consider performing additional experiments in specifically differentiated cells. However, the method we employed is still the most used, and mostly gives rise to CMs with a ventricular-like phenotype.

Regarding the further considerations of the reviewer on the depolarized phenotype of our cells and the consequences on the shown data, this is another subject already extensively discussed in the manuscript since the first submission. With the intent to prove that differences between CNTR- and K219T-CMs on the dV/dt_{max} parameter were not due to the different polarization of the cells, we performed a series of experiments on CMs (both CNTR and K219T) electrically forced to -82mV. Results from these experiments confirmed those obtained from the spontaneously beating CM populations (Supplementary Table 2 and 3).

However, in order to avoid misunderstanding and to clarify the approach we used for the functional analysis of the CMs, we have added in the supplementary material (see Extended Methods Section) a paragraph on the validation of the electrophysiological interactions entitled "Assessment of MDP and dV/dt_{max} impact on I_{Na} derived from different populations of LMNA-CM AP phenotypes: further electrophysiological considerations". We hope this piece of information will make the whole approach clearer to the reviewer and to future readers.

4) The authors go to great length to correct the K219T mutation using CRISPR/Cas9 mediated genome editing and test to see how the correction affects SCN5A gene expression and sodium current density. Yet surprisingly, no functional AP data at the single cell level (Figure 1) nor conduction data at the multicellular level (Figure 3) is presented for the corrected line.

To confirm the authors' conclusions, it will be necessary to test the direct effects of the correction on conduction velocity, AP parameters, and pro-arrhythmic activity in the genome-edited corrected line. This data should than be accordingly compared to data acquired from both control and K219T lines.

We are sure the reviewer is aware of the enormous efforts we made in generating gene-corrected lines and to perform the key experiments to demonstrate the causal link between the K219T mutation and

the phenotype. The request of generating gene-corrected lines dates back to the first revision and, due to time constraints, we chose at that time to analyse CMs from gene-corrected lines for the main Na⁺ current phenotype, assessing whether Na⁺ current density was re-established in absence of the mutation, and to validate the proposed molecular mechanisms, performing gene expression and ChIP experiments on *SCN5A*.

From the reviewer's response to the second version we submitted, we understood he/she was satisfied with those experiments, at least for the functional part. Indeed, the reviewer commented on this part of the manuscript as follow: "*I appreciate the authors' efforts to address my previous comments, especially generation of additional iPSC lines (Fig. 7). The current revised manuscript showed increase quality and quantity of data that support below two main points of this manuscript. I have no additional questions for these main points:*

- 1) *Lamin A/C mutation (K219T) induces abnormal sodium current of mutant iPSC-CMs.*
- 2) *Decreased SCN5A is required for disease phenotype in mutant iPSC-CMs*".

Apparently, the reviewer has now changed his/her mind on this point; this is unacceptable, regardless of the fact that data from the requested experiments would not give any additional value to the manuscript, since genotype-phenotype correlations have been established already at the functional and molecular levels.

5) *The authors described in rebuttal letter that "second, the two genes sharing the LAD domain with SCN5A (namely, SCN10A and SCN11A) are not expressed in the CMs, neither they are modulated between the two conditions, but rather they are neuronal isoforms of the sodium channel Nav1.8 and Nav1.9 (Verkek A.O. et al, 2012, DibHaji S.D. et al., 2015, Marban E. et al, 1998 and unpublished RNAseq data by us see in Figure 1 of this reply to reviewers)".*

How did authors check SCN5A, SCN10A and SCN11A are located in same LAD? Did authors utilize published data set for LMNA (or LMNB1 ChIPseq)? In LMNA-related research, most researchers have tested the concept of "LAD (Lamin associated domain)" to explain their hypothesis (J Cell Biol. 2015 Jan 5;208(1):3352) (J Cell Biol. 2017 Sep 4;216(9):27312743) (Hum Mol Genet. 2018 Apr 15;27(8):14471459). And a recent paper showed how LAD correlated with H3K9me2 histone modification (Cell. 2017 Oct 19;171(3):573587) that is closely related with PRC2 and H3K27me3 histone maker. I think the authors really have to seriously consider and reconcile their findings and hypothesis with these other papers.

We do not understand the link between the question raised by the reviewer and the content of the manuscript. Discussion on the two genes *SCN10A* and *SCN11A* were started by the reviewer in the third revision, where he/she specifically asked to check the enrichment of PRC on the genes that share the LAD with *SCN5A* ("*...Authors should check the enrichment of PRC on the genes that share LAD with SCN5A*"...).

Since we did not have any ChIP-sequencing data on our iPSC-CMs, we based our reply on other papers that identify, albeit in other cell types, *SCN11A-SCN10A* and *SCN5A* as included in a LAD through genome-wide analysis (Lund et al., 2013; Lund et al., 2015). Also, our answer was limited in saying that *SCN10A* and *SCN11A* were not relevant to the cardiac physiology in question, neither resulting to be modulated in our system; therefore, answering this question was outside the scope of the manuscript. Similarly, it is unclear why we should reconsider our hypothesis and findings based on the suggested literature.

As a matter of fact, assessing the effect of the K219T mutation on LAD gain/loss was not the goal of our study, since we agree with the reviewer that answering that specific question will require to determine the global occupancy of Lamin A/C in the entire genome. The hypothesis of an alteration of LADs in mutant CMs (in particular through cardiac differentiation) raised in the manuscript was purely speculative and was suggested by reviewer #1; this hypothesis is in agreement with most of the literature cited by the reviewer (Briand et al., 2018; Oldenburg et al., 2017). Furthermore, results

from Poleshko et al. (Cell. 2017 Oct 19;171(3):573587), are also supporting a role of the nuclear lamina (Lamin B in this case) in modulating gene expression and specifically show a shift of cardiac-specific genes from the nuclear periphery to the nucleoplasm during cardiac differentiation of mouse ESCs. In the text, we just mentioned the possibility that *SCN5A* could be included in a LAD, but again, this is not the main scope of the study. To avoid misunderstanding, in this new version we have nonetheless reconsidered our speculations and clearly indicated that only genome-wide data may firmly prove that LAD remodelling is occurring.